# Household alternating current electricity plug-and-play quantum-dot light-emitting diodes

Jiming Wang[1,2], Cuixia Yuan[1] & Shuming Chen [1] ✉

As an intrinsically direct current device, quantum-dot LED cannot be directly driven by household alternating current electricity. Thus, a driver circuit is required, which increases the complexity and cost. Here, by using a transparent and conductive indium-zinc-oxide as an intermediate electrode, we develop a tandem quantum-dot LED that can be operated at both negative and positive alternating current cycles with an external quantum efficiency of 20.09% and 21.15%, respectively. Furthermore, by connecting multiple tandem devices in series, the panel can be directly driven by household alternating current electricity without the need for complicated back-end circuits. Under 220 V/50 Hz driving, the red plug-and-play panel demonstrates a power efficiency of 15.70 lm W$^{-1}$ and a tunable brightness of up to 25,834 cd m$^{-2}$. The developed plug-and-play quantum-dot LED panel could enable the production of cost-effective, compact, efficient, and stable solid-state light sources that can be directly powered by household alternating current electricity.

Light-emitting diodes (LEDs) have become the mainstream lighting technology due to their advantages of high efficiency, exceptional longevity, solid state, and environmental safety, which fulfills the global demand for energy efficiency and environmental sustainability[1–4]. As a semiconductor pn diode, LED can only be operated under the driving of a low-voltage direct current (DC) source. Due to the unidirectional and continuous charge injection, the charges as well as the Joule heat can accumulate within the devices, thereby reducing the operational stability of LEDs[5,6]. Moreover, the global electricity supply is dominated by high-voltage alternating current (AC), which cannot be directly used by numerous household appliances like LED lamps. Therefore, when LEDs are driven by household electricity, an additional AC-DC converter is required as an intermediary to convert high-voltage AC into low-voltage DC. Typical AC-DC converters consist of a transformer to step down the mains voltage and a rectification circuit to rectify the AC input (Supplementary Fig. 1a). Although most AC-DC converters can achieve conversion efficiencies of over 90%[7–11], energy losses remain during the conversion process. In addition, to tune the brightness of LEDs, a dedicated driver circuit should be used to regulate the DC power and supply the ideal current to the LEDs

(Supplementary Fig. 1b)[12,13]. The reliability of driver circuits can impact the durability of LED lamps. As a result, introducing the AC-DC converters and DC drivers not only incurs additional costs (accounting for approximately 17% of the entire LED lamp cost)[14] but also increases power consumption and reduces the robustness of LED lamps. Therefore, it is highly desirable to develop LEDs or electroluminescence (EL) devices that can be directly driven by household 110 V/220 V at 50 Hz/60 Hz without complicated back-end electronics.

Over the past few decades, several AC-driven EL (AC-EL) devices have been demonstrated[15–50]. Typical AC-EL devices consist of a phosphor emissive layer sandwiched between two insulating layers (Supplementary Fig. 2a)[15–26]. The use of insulating layers blocks the injection of external charge carriers and thus there is no DC current flowing through the devices. The device functions as a capacitor and under the driving of a high AC electric field, the electrons, internally generated from the trapped sites, can tunnel into the emissive layer. After gaining sufficient kinetic energy, the electrons impact the luminescent centers, thereby leading to the generation of excitons and light emission[15–18]. Due to the inability to externally inject electrons from the electrodes, these devices demonstrate notably low levels of

[1]Department of Electrical and Electronic Engineering, Southern University of Science and Technology, Shenzhen 518055, PR China. [2]Harbin Institute of Technology, Harbin 150001, PR China. ✉e-mail: chen.sm@sustech.edu.cn

brightness and efficiency, thus limiting their suitability for applications in lighting and displays[15–24]. In efforts to enhance their performance, AC-EL devices featuring a single insulating layer have been devised (Supplementary Fig. 2b)[27–40]. In this structure, during the positive half cycles of AC driving, one type of charge carrier is directly injected from the external electrodes into the emissive layer; by recombining with the other type of charge carriers that are internally generated, efficient light emission can be observed. However, during the negative half cycles of AC driving, the injected charge carriers are released from the devices, and thus there is no light emission. Since light emission occurs only in a half cycle of driving, such AC devices are less efficient than DC devices. Moreover, due to the capacitive nature of the devices, both types of AC devices exhibit frequency-dependent EL performance[28–34], and the best performance is usually achieved at a high frequency of several kHz[18–23,27–40], making them challenging to be compatible with a low frequency (50 Hz/60 Hz) of standard household AC electricity.

Recently, an AC-EL unit that can be operated at 50 Hz/60 Hz has been proposed[51–60]. Such a unit consists of two side-by-side DC devices (Supplementary Fig. 2c)[54–59]. By electrically shorting the top electrodes of two devices, and connecting the bottom coplanar electrodes to an AC source, the two devices can be turned on alternately. From the circuitry perspective, such an AC-EL unit is obtained by serially connecting a forward device and an inverted device. When the forward device is turned on, the inverted device is switched off and functions as a resistor. Due to the presence of resistors, the EL efficiency is relatively low[56–58]. In addition, the AC-EL unit can only be operated at low voltage, which cannot be directly integrated into 110 V/220 V standard household electricity. As summarized in Supplementary Fig. 3 and Supplementary Table 1, the reported AC-EL devices driven by high AC voltage exhibit lower performance (brightness and power efficiency) than that of the DC devices, and thus far, there is no AC-EL device that can be directly driven by 110 V/220 V at 50 Hz/60 Hz household electricity with high efficiency and long lifetime.

In this contribution, we present tandem quantum-dot LEDs (QLEDs) capable of direct operation using standard household electricity of 110 V/220 V at 50 Hz/60 Hz. The QLEDs exhibit remarkable features, including tunable emission color, high color saturation, elevated brightness levels, and simple solution processability. Consequently, they have garnered significant attention as prospective contenders for both light sources and displays in recent research endeavors[61–67]. Our developed tandem device consists of two vertically stacking QLEDs that are connected by a transparent and conductive indium-zinc-oxide (IZO) intermediate electrode. By shorting the bottom and top electrodes as a common electrode, and extracting the IZO intermediate electrode as a counter electrode, two QLEDs with opposite polarity are connected in parallel. The resulting tandem device can be operated at both negative and positive cycles of the AC voltage with a high external quantum efficiency (EQE) of 20.09% and 21.15%, respectively. Under AC driving, the accumulated charges and Joule heat can be effectively reduced, and thus the lifetime of QLEDs is significantly prolonged. Furthermore, by connecting multiple tandem QLEDs in series, a household AC electricity plug-and-play QLED (PnP-QLED) light source is obtained. Under 220 V/50 Hz driving, the (PnP-QLED)$_{30}$ demonstrates a high power efficiency (PE) of 15.70 lm W$^{-1}$ and a tunable brightness of up to 25,834 cd m$^{-2}$. The developed (PnP-QLED)$_n$ ($n$ is the number of PnP-QLEDs used for connection) could enable the production of low-cost, compact, efficient, and stable solid-state light sources that can be directly powered by household AC power.

## Results
### Principle and performance of tandem QLEDs
As schematically shown in Fig. 1a, our proposed tandem device consists of a bottom (B) QLED and a top (T) QLED that are vertically stacked and connected by a transparent and conductive IZO

intermediate electrode. The use of IZO plays a key role, as it enables the tandem device to operate in both DC and AC modes. Under the DC mode, the tandem device is denoted as S-QLED, where both B-QLED and T-QLED are connected in series. Driven by a DC source, both B-QLED and T-QLED are turned on simultaneously. Under the AC mode, the tandem device is denoted as AC-QLED, where the top Al of T-QLED and bottom indium-tin-oxide (ITO) of B-QLED are electrically shorted to serve as a common electrode, while the IZO is extracted as a counter electrode. In this configuration, the B-QLED and T-QLED with opposite polarity are connected in parallel, as schematically shown in Fig. 1a. Compared to the reported AC-EL devices, the advantages of our AC-QLED are as follows. Firstly, our AC-QLEDs ensure continuous light emission. Under the driving of an AC source, the B-QLED and T-QLED are alternately turned on, thereby allowing light emission to be continuously observed in either the positive or negative cycles of AC voltage. Secondly, our devices show high performance and longer lifetime. Even driven by an AC source, the QLEDs inherently operate in DC mode when they are turned on, allowing them to achieve high brightness and efficiency. Besides, a longer lifetime is expected due to the introduction of a negative bias when they are switched off. Lastly, our QLEDs maintain superior performance even at low frequency of 50 Hz/60 Hz, owing to the fact that they are not capacitive devices. The above advantages make our tandem QLED a very promising candidate for realizing efficient household AC electricity plug-and-play (PnP) light sources.

The optical performance of the devices can be simulated by using a dipole radiation model (Supplementary Fig. 4)[68–71]. The simulation methods and refractive index (Supplementary Figs. 4–6) used for simulation are detailed in the Supplementary Simulation Method. To validate the accuracy of the simulation, we compare the simulated EQE/$\gamma$ with the measured EQE, where $\gamma$ is the electrical efficiency and is determined by the ratio of generated excitons to the injected hole-electron pairs. As shown in Supplementary Fig. 7h, the trend of the EQE/$\gamma$ of the QLED is in good agreement with the measured EQE, confirming the accuracy and reliability of our simulated methods. Optically, the tandem device can be modeled as two vertically stacked Fabry-Pérot resonant cavities, as shown in Fig. 1b. The length of one cavity is affected by the length of another cavity. Therefore, to achieve constructive wide-angle interference and enhance the light out-coupling efficiencies (OCEs) of both B-QLED and T-QLED, the length of both cavities should be simultaneously optimized. By tuning the thickness of the bottom ITO and ZnMgO (ZMO_B), it is possible to optimize the OCEs of both B-QLED and T-QLED simultaneously without compromising their electrical performance. As shown in Supplementary Fig. 8, with 80 nm ITO and 110 nm ZMO_B, both B-QLED and T-QLED exhibit the highest OCEs of 29.3% and 30.5%, respectively. Similarly, the OCEs of both B-QLED and T-QLED are influenced by the thickness of the intermediate IZO electrode and the top ZnMgO (ZMO_T) layer. As shown in Supplementary Fig. 9 and Fig. 1c, the optimal thicknesses of IZO and ZMO_T are 30 nm and 90 nm, respectively, which result in the highest OCEs of over 30% for both B-QLED and T-QLED and over 61.5% for the S-QLED.

Guided by the simulation results, tandem devices with optimal thickness were fabricated. The B-QLED (with an optimal structure of glass/ITO 80 nm/ZMO_B 110 nm/QD 20 nm/CBP 50 nm/MoO$_3$ 10 nm/ IZO 30 nm) was prepared first, followed by spin-casting the ZMO_T on top of the IZO. During the fabrication of the ZMO_T (Supplementary Fig. 10a), the solvent ethanol can penetrate the B-QLED and cause damage to the organic layers (Supplementary Fig. 10c), leading to the cracking of the IZO (Fig. 1d and Supplementary Fig. 10e). To tackle this issue, a dynamic spin-casting method (Supplementary Fig. 10b) was developed to deposit the ZMO_T, which can effectively mitigate the solvent-induced damage and thereby maintain the integrity of the IZO (Fig. 1e, Supplementary Fig. 10d, f). The developed fabrication processes (Supplementary Fig. 11) allow us to realize damage-free tandem

devices, as evidenced by the uniform light emission from B-QLED, T-QLED, and S-QLED, respectively (Fig. 1f–h).

By extracting the IZO as an independent electrode, the performance of both B-QLED and T-QLED can be individually evaluated. As shown in Fig. 1i and Table 1, with an optimal IZO thickness of 30 nm, the B-QLED and T-QLED exhibit an EQE of 20.09% and 21.51%, respectively, which is very close to that of the single device (Supplementary Fig. 7), indicating that the optical and electrical performances of the tandem

device are fully optimized. As shown in Fig. 1i, j, and Supplementary Fig. 12, the current density-voltage-luminance (*J-V-L*) and EQE-*J* characteristics of the S-QLED (tandem device operates in DC mode) are perfectly equal to the summation of those of B-QLED and T-QLED, indicating there are no optical or electrical performance losses in the tandem device. Such a result also implies that both B-QLED and T-QLED are effectively connected by the IZO electrode. The key performances of the devices are summarized in Table 1.

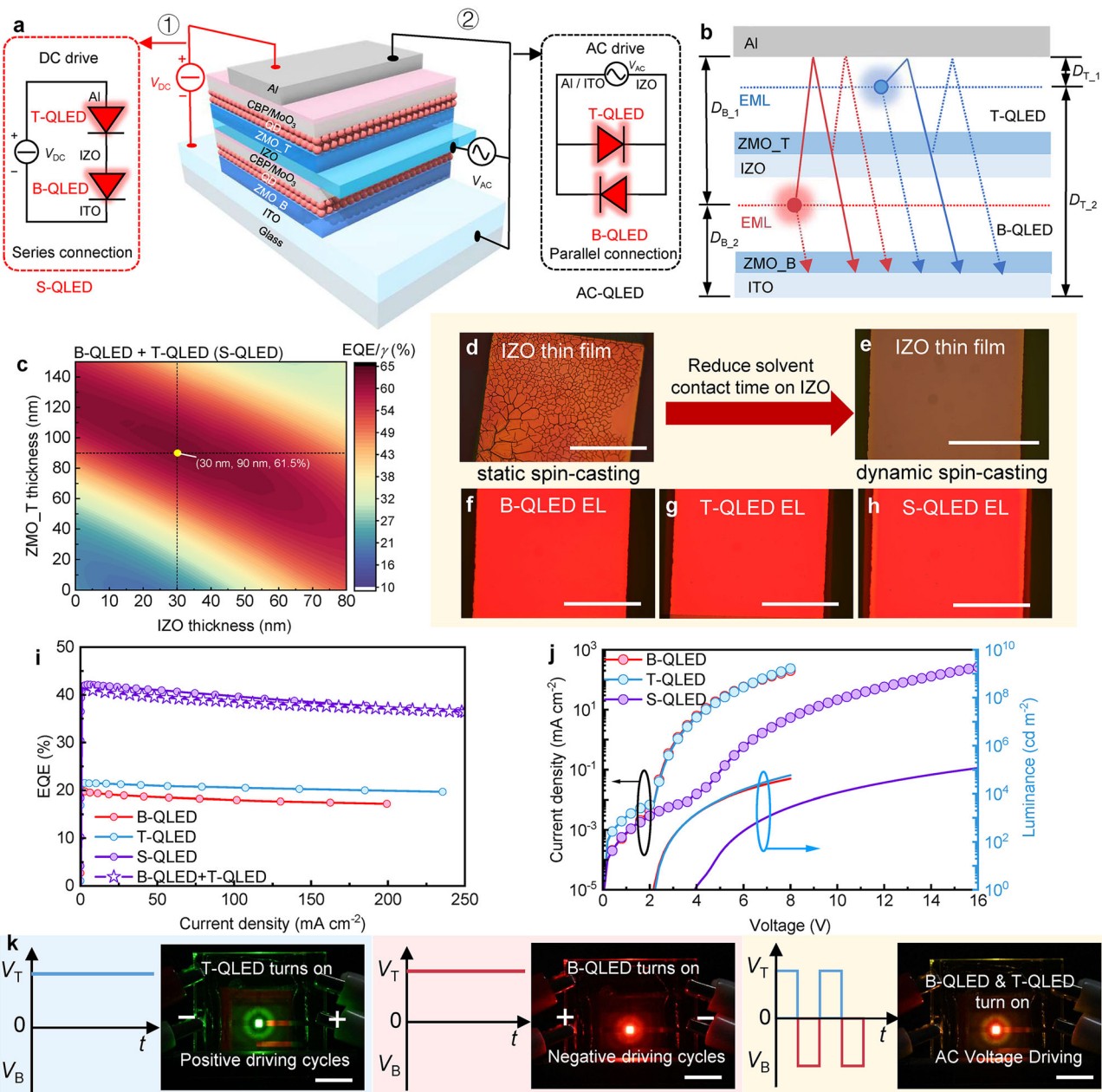

**Fig. 1 | Principle and performance of tandem QLED. a** Device structure. The tandem device consists of two vertically stacking QLEDs that are connected by an IZO intermediate electrode (ZMO_B/ZMO_T represents the bottom/top ZnMgO layer). The use of IZO enables the tandem device to operate in either direct current (DC) or alternating current (AC) mode ($V_{DC}$/$V_{AC}$ represents the direct/alternating current voltage). **b** Schematic optical model of a tandem QLED (EML represents emission layer). To achieve constructive wide-angle interference, the length of both cavities should be simultaneously optimized ($D_{B\_1}$/$D_{T\_1}$ and $D_{B\_2}$/$D_{T\_2}$ are the distances of the bottom/top recombination zone from the top Al electrode and the bottom ITO electrode, respectively; B-QLED/T-QLED represents the bottom/top QLED in tandem QLED). **c** Simulated outcoupling efficiencies (OCEs) of both

B-QLED and T-QLED as a function of the thickness of ZMO_T and IZO ($\gamma$: charge balance efficiency). **d–h** The optical images of the surface of IZO thin films after static spin-casting (**d**), and dynamic spin-casting with ethanol solvent (**e**), as well as the electroluminescence (EL) images of B-QLED (**f**), T-QLED (**g**), and tandem QLED (denoted as S-QLED, where both B-QLED and T-QLED are connected in series) (**h**). Scale bars, 1 mm. The *J-V-L* (**i**), and EQE-*J* characteristics (**j**), of B-QLED (red solid circle), T-QLED (blue solid circle), and S-QLED (purple solid circle), respectively: the EQE of the S-QLED (purple solid circle) are perfectly equal to the summation of those of B-QLED and T-QLED (purple open star). **k** The photos of a tandem device that was operated in alternating current mode at different driving conditions (also demonstrated in Supplementary Movie 1). Scale bars, 5 mm.

**Table 1 | The performances of a red tandem QLED**

| Device | $V_{on}$ (V) | CE (cd A$^{-1}$) | | EQE (%) | | PE (lm W$^{-1}$) | |
|---|---|---|---|---|---|---|---|
| | | 1000/ 10,000 (cd m$^{-2}$) | max | 1000/ 10,000 (cd m$^{-2}$) | max | 1000/ 10,000 (cd m$^{-2}$) | max |
| B-QLED | 1.80 | 24.17/ 22.89 | 24.79 | 19.63/ 18.60 | 20.09 | 19.97/ 12.39 | 29.94 |
| T-QLED | 2.00 | 26.99/ 26.55 | 27.07 | 21.46/ 21.10 | 21.51 | 22.31/ 14.89 | 30.13 |
| S-QLED | 3.40 | 51.24/ 51.07 | 51.58 | 41.88/ 41.75 | 42.16 | 22.98/ 16.04 | 26.16 |

Figure 1k shows the photos of a tandem device that was operated in AC mode. To distinguish the emission color, a red B-QLED and a green T-QLED were used. Under the driving of a positive voltage, only the green T-QLED is turned on, while under the driving of a negative voltage, only the red B-QLED is activated. Therefore, driven by an AC source, both B-QLED and T-QLED are alternately turned on, as demonstrated in Supplementary Movie 1. The results indicate that our tandem device can perfectly operate in AC mode.

## Performance of alternating current-driven QLEDs
The equivalent circuit of the tandem QLED operated in AC mode is shown in Fig. 2a, b. In the positive driving cycles (Fig. 2a), the T-QLED is

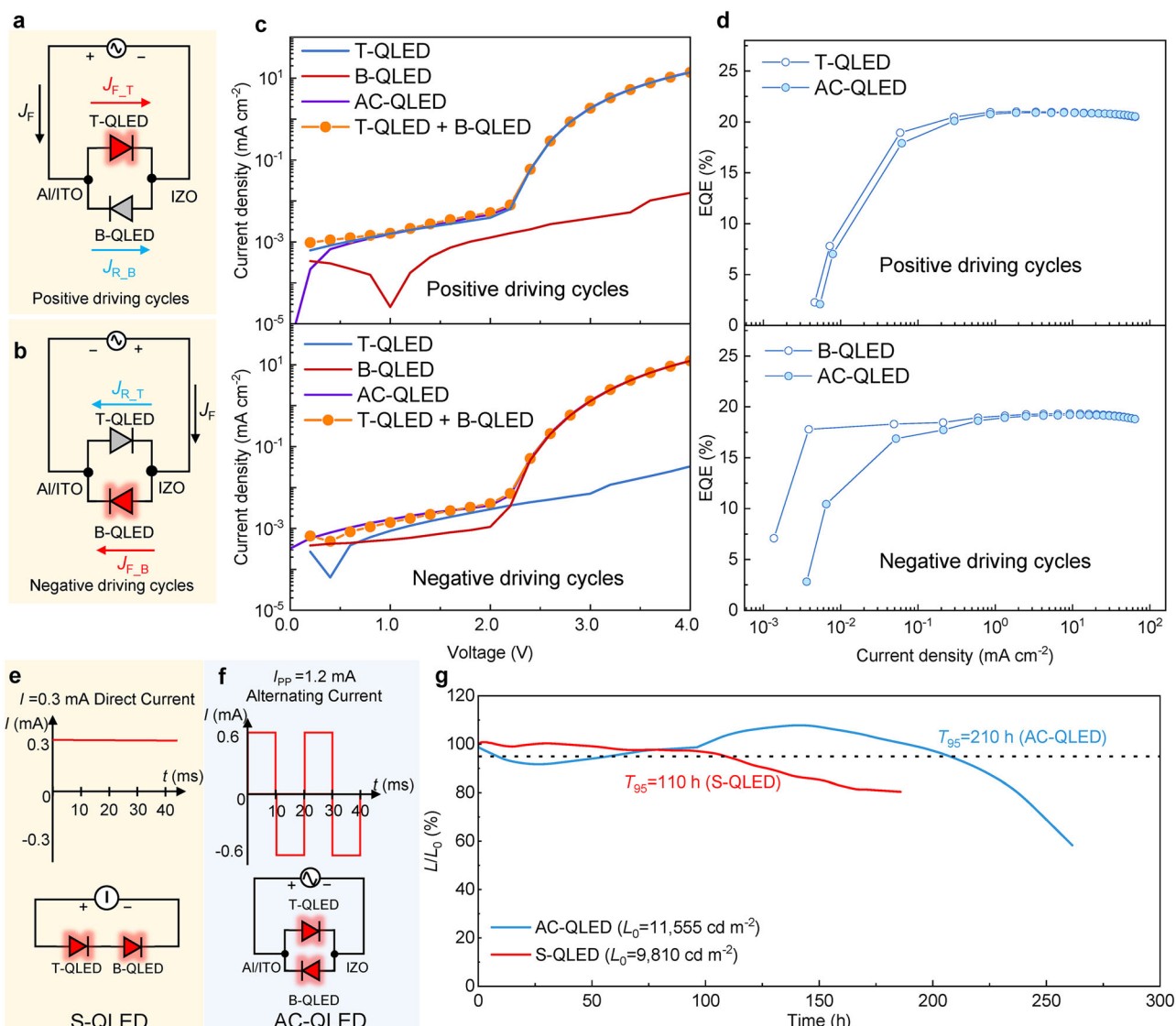

**Fig. 2 | Performance of AC-QLED. a, b** Equivalent circuit of the AC-QLED in positive driving cycles ($J_{F\_T}/J_{R\_B}/J_F$: forward/reverse/total current density from the T-QLED/B-QLED/AC-QLED) (**a**), and negative driving cycles ($J_{F\_B}/J_{R\_T}/J_F$: forward/reverse/total current density from the B-QLED/T-QLED/AC-QLED) (**b**). B-QLED/T-QLED represents the bottom/top QLED in tandem QLED, and AC-QLED represents a tandem QLED that was operated in alternating current mode. **c** The $J$-$V$ characteristics of T-QLED (blue curve), B-QLED (red curve), and AC-QLED (purple curve) in positive driving cycles and negative driving cycles: the $J_F$ (total current density in positive/ negative driving cycles, purple curve) of the AC-QLED are perfectly equal to the summation of those of B-QLED and T-QLED (orange solid circle). **d** The EQE-$J$ characteristics of T-QLED (open circle), B-QLED (open circle), and AC-QLED (solid circle) in positive driving cycles and negative driving cycles, respectively: the significant decrease in EQE of AC-QLED is evidently caused by $J_{R\_B}/J_{R\_T}$ in positive/ negative driving cycles. **e, f** The driving conditions for S-QLED (a tandem QLED was operated in direct current mode) (**e**), and AC-QLED (**f**), for lifetime testing. The S-QLED is powered at 0.3 mA direct current ($I$), and the AC-QLED is powered at 50 Hz and 1.2 mA peak-to-peak alternating current ($I_{PP}$). **g** The lifetime curves of the S-QLED (red curve) and AC-QLED (blue curve). The initial luminance ($L_0$) of S-QLED and AC-QLED are 9810 cd m$^{-2}$ and 10,000 cd m$^{-2}$, respectively ($L$ represents the luminance of the device during the testing process).

turned on, while the B-QLED is switched off. Thus, the total current density ($J_F$) in positive driving cycles is equal to the summation of forward current density of T-QLED ($J_{F\_T}$) and the reverse current density of B-QLED ($J_{R\_B}$). A small $J_{R\_B}$ is preferred since it does not contribute to photon generation. By individually driving the T-QLED and B-QLED, the $J_{F\_T}$ and $J_{R\_B}$ can be measured. As shown in Fig. 2c, the $J_{R\_B}$ is several orders of magnitude lower than the $J_{F\_T}$, and thus the total current $J_F$ is mainly determined by $J_{F\_T}$. As a result, the EQE of AC-QLED in the positive driving cycles is almost equal to that of T-QLED, especially in the high current range where the $J_{R\_B}$ can be neglected. Similar results are also obtained when the AC-QLED operates in the negative driving cycles (Fig. 2b, c).

At low current or low brightness levels, the reverse current cannot be neglected, as it greatly affects the efficiency losses of the AC-QLED. As shown in Fig. 2d, it can be observed that the efficiency losses of AC-QLED under positive and negative driving cycles are not equal. Due to the rougher surface induced by the thick bottom layers, the reverse current density $J_{R\_T}$ (from the T-QLED) during the negative driving cycles is much higher than the reverse current density $J_{R\_B}$ (from the B-QLED) during the positive driving cycles, as illustrated in Fig. 2c. Consequently, under negative driving cycles, the energy losses in the AC-QLED are higher compared to those under negative driving cycles. However, for high currents or high brightness situations, because the forward current is significantly higher than the reverse current, the impact of the reverse current on the efficiency of AC-QLED becomes negligible. Due to the low reverse current of the diode, the reversely biased devices do not incur appreciable energy losses to the AC-QLED. For example, at a brightness of 1000 cd m$^{-2}$, the AC-QLED exhibits a PE of 26.46 lm W$^{-1}$ in positive driving cycles and 20.99 lm W$^{-1}$ in negative driving cycles (Supplementary Fig. 13), which is close to 26.59 lm W$^{-1}$ and 21.17 lm W$^{-1}$ of the T-QLED and B-QLED, respectively. As illustrated in Supplementary Fig. 14, the energy loss of the AC-QLED accounts for less than 1% under high voltage or luminance, significantly lower than the energy loss in switching power supplies (AC-DC converters and DC-DC driver circuits), which typically ranges from 10% to 20%[9–13]. Consequently, driven by a 50 Hz square wave AC voltage, the red AC-QLED exhibits an average PE of 23.16 lm W$^{-1}$ at an average brightness of 1000 cd m$^{-2}$, which are the best ever reported for AC-EL devices, as compared in Supplementary Table 1 and Supplementary Fig. 3.

Although the efficiency of AC-QLED is almost the same as that of QLED driven by a DC source, its lifetime is effectively prolonged. During forward driving, the charge carriers are continuously injected and accumulated in the devices, leading to the rise of the driving voltage, the quenching of light emission, and eventually the degradation of devices[49]. By applying a negative bias, the accumulated charges can be effectively released, resulting in a partial recovery of the luminescence, as demonstrated in Supplementary Fig. 15. Therefore, by operating the tandem device in AC mode, the $T_{95}$ lifetime (time for the brightness decreasing to 95% of its initial value of 10,000 cd m$^{-2}$) reaches 216 h, which is almost two times longer than 110 h of the tandem device operated in DC mode, as shown in Fig. 2e–g.

### The basic plug-and-play QLED

The developed AC-QLED can be operated over a wide frequency range of several Hz-MHz. However, its driving voltage is relatively low, making it difficult to directly integrate into high-voltage AC electricity. By connecting many AC-QLEDs in series, the driving voltage can be multiplied. The resulting light source can be regarded as a PnP device as it is compatible with any AC voltage and frequency. A basic PnP-QLED consists of two side-by-side AC-QLEDs that are connected in series by the intermediate IZO electrode, as shown in Fig. 3a. The fabrication processes of the PnP devices are shown in Supplementary Fig. 16. In the positive driving cycles, the T-QLED from the first tandem

device and the B-QLED from the second tandem device are turned on, while in the negative driving cycles, the B-QLED from the first tandem device and the T-QLED from the second tandem device are activated, as shown in Fig. 3b and demonstrated in Supplementary Movie 2. No matter what the driving cycles are, light emission always originates from a T-QLED and a B-QLED, thereby ensuring that the PnP-QLED outputs a constant brightness throughout the entire AC driving. This addresses the issue of unequal efficiency between B-QLED and T-QLED during AC driving. The optical performance of PnP-QLED is close to that of S-QLED because for these two types of devices, light emission always originates from both T-QLED and B-QLED. However, the electrical performance is different. As shown in Fig. 3c, compared to S-QLED, the PnP-QLED shows a higher leakage current (region A), which is due to the presence of reverse current in the parallel circuit. Fortunately, the reverse current is relatively small and only slightly affects the EQE in the low brightness region (Fig. 3d). As the driving voltage is increased, the PnP-QLED exhibits lower current (region B) and luminance, which is induced by the resistance of IZO wire. By calculating the power difference ($\Delta P$) between the PnP-QLEDs with a 0.2 mm and a 0.5 mm long IZO wire (Supplementary Fig. 17), and fitting $\Delta P$ with $\triangle P = I^2 R_{IZO}$, where $I$ is the current in the PnP-QLED and $R_{IZO}$ is the resistance of IZO wire, the $R_{IZO}$ can be calculated as 0.2 Ω μm$^{-1}$. Despite the presence of $R_{IZO}$ and reverse current, the PnP-QLED still exhibits a peak PE of 20.46 lm W$^{-1}$, which is slightly lower than 22.66 lm W$^{-1}$ of the S-QLED, as shown in Fig. 3e. By replacing the IZO with a more conductive wire, the PE can be further improved. Furthermore, as shown in Supplementary Fig. 18, the $T_{95}$ lifetime (initial average luminance of 10,020 cd m$^{-2}$) of the PnP-QLED under sinusoidal AC driving reaches 160 h, which is 1.5-fold longer than that of the PnP-QLED driven by a DC source. The results confirm that the PnP-QLED is capable of achieving both high efficiency and prolonged lifetime when operated under household AC electricity.

### Household alternating electricity-driven plug-and-play QLED

Using the basic PnP-QLED as a building block, a series of light sources that can be directly plugged into household AC electricity can be realized. Figure 4a shows the schematic circuit diagram of several PnP-QLEDs that are connected in series. The resulting light source is labeled as (PnP-QLED)$_n$, where $n$ denotes the number of PnP-QLEDs used for connection. Except for the need to pattern the bottom ITO, the intermediate IZO, and the top Al electrodes, all other fabrication processes are the same as for regular QLEDs, as shown in Supplementary Fig. 19a. The PnP-QLEDs are connected by metallic Al (Supplementary Fig. 19b), which is highly conductive, and thus the power consumption caused by Al wires is negligible. Hence, no matter how many PnP-QLEDs are connected, the PE of (PnP-QLED)$_n$ is the same as that of the basic PnP-QLED. By varying the $n$, the operational voltage of (PnP-QLED)$_n$ can be tuned to match arbitrary AC voltage levels.

To demonstrate that the (PnP-QLED)$_n$ can be directly driven by household 220 V/50 Hz AC electricity, the $n$ is varied. To realize the maximum PE, an ideal $n$ should be used. The average voltage dropped across each PnP-QLED is 220/$n$ V; by finding the corresponding PE at that voltage from the PE-$V$ curve of PnP-QLED (Supplementary Fig. 20), the PE of (PnP-QLED)$_n$ can be obtained. The calculated PE as a function of $n$ is shown in Fig. 4b. With an ideal $n = 37$ or 38, a maximum PE of 20.46 lm W$^{-1}$ can be achieved. However, due to sample size limitations, we could only connect a maximum of 30 PnP-QLEDs. With our custom measurement setup (Supplementary Fig. 21), the AC voltage, current, and luminance of the (PnP-QLED)$_n$ can be measured (Supplementary Figs. 22–25d). Supplementary Fig. 22d presents the AC voltage and current of the (PnP-QLED)$_{30}$ under 220 V/50 Hz driving. The average current ($I_{avg}$) and power

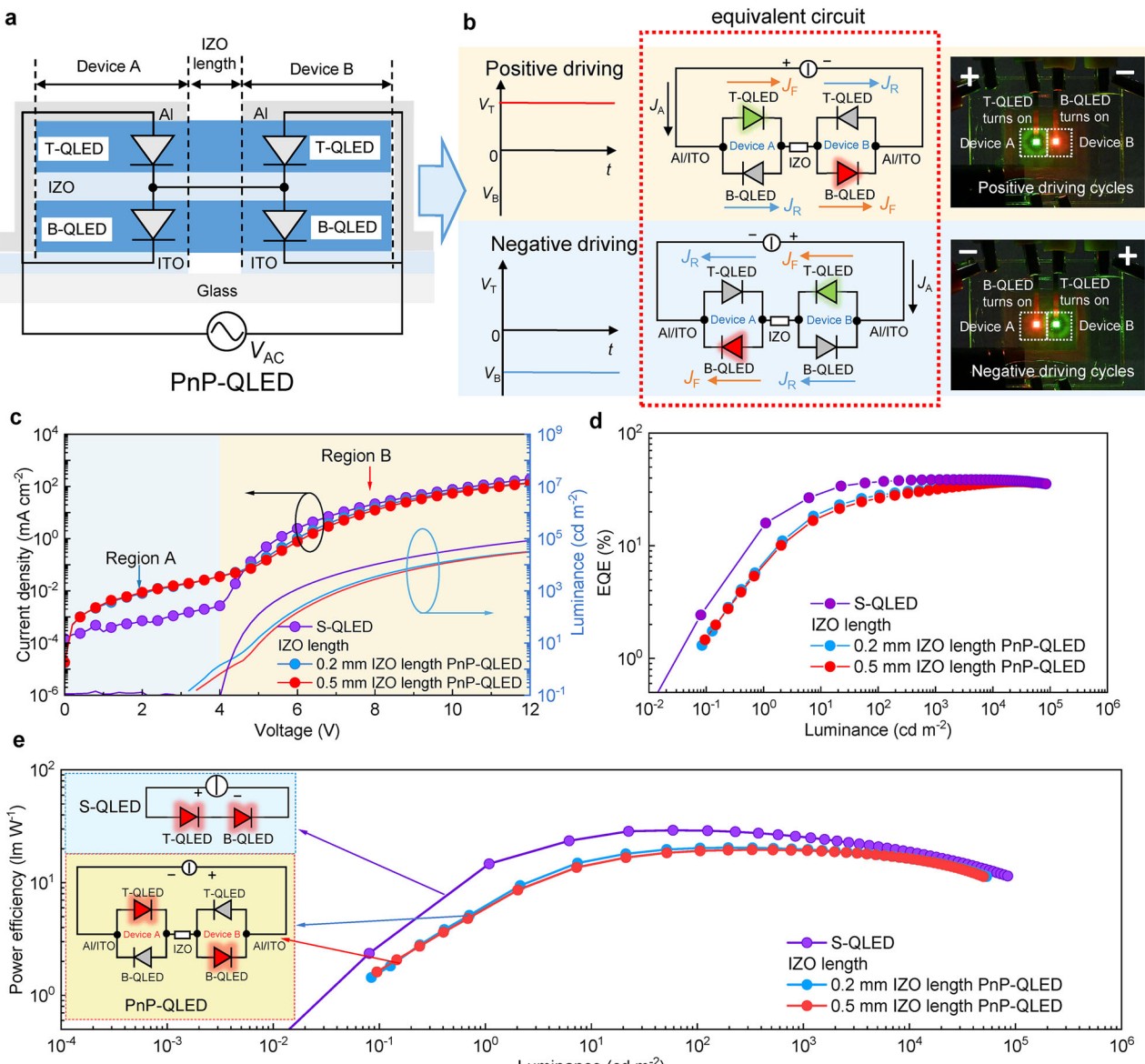

**Fig. 3 | Operating principle and performance of PnP-QLED. a** Schematic device structure of a plug-and-play QLED (PnP-QLED). A basis PnP-QLED consists of two side-by-side AC-QLEDs connected in series. By doing so, the driving voltage is doubled. B-QLED/T-QLED represents the bottom/top QLED in tandem QLED. AC-QLED represents a tandem QLED that was operated in alternating current mode. **b** Schematic circuit diagrams and photos of a PnP-QLED under the positive and negative driving cycles: $J_F/J_R$ represents the forward/reverse current density in PnP-

QLED (also demonstrated in Supplementary Movie 2). The *J-V-L* (**c**), EQE-*L* (**d**), and PE-*L* characteristics (**e**), of PnP-QLEDs (the blue/red circle represents the PnP-QLED with 0.2 mm/0.5 mm lengths of IZO wire) and S-QLED (purple circle), respectively. S-QLED represents a tandem QLED that was operated in direct current mode. The significant decrease in EQE and PE of PnP-QLED is evidently induced by the reverse current in the parallel circuit and resistance of IZO wire.

($P_{avg}$) consumed of the (PnP-QLED)$_n$ can be calculated by:

$$I_{avg} = \frac{1}{T}\int_0^T I_{RMS}(t)dt \tag{1}$$

$$P_{avg} = \frac{1}{T}\int_0^T V_{RMS}(t)I_{RMS}(t)dt \tag{2}$$

where $T$ is the period of AC driving, $V_{RMS}(t)$ and $I_{RMS}(t)$ are the root mean square voltage and current. Additionally, a photodiode was used to measure the average radiated luminous flux (lumens) of (PnP-QLED)$_n$ over multiple periods. With the $I_{avg}$, $P_{avg}$, and lumens as the inputs, the PE, EQE, and CE of the (PnP-QLED)$_n$ can be obtained.

As shown in Fig. 4b, by increasing the $n$ from 26 to 30, the PE is gradually enhanced from 8.03 lm W$^{-1}$ to 12.70 lm W$^{-1}$. Such a trend is consistent with the calculation. The lower PE is due to the poor uniformity induced by the large size (over 1 cm$^2$) of samples. By improving the laboratory fabrication conditions, a theoretically high PE of 20.46 lm W$^{-1}$, which is similar to that of PnP-QLED, can be obtained.

Figure 4c shows the PE of (PnP-QLED)$_{30}$ as a function of AC voltage. The PE is gradually decreased as the AC voltage is increased, which is consistent with the calculation and follows the same trend as conventional QLED. The PE, CE, and EQE of the (PnP-QLED)$_{26-30}$ are also shown in Supplementary Fig. 25a–c. By varying the AC voltage, the brightness can be effectively tuned from 1000 cd m$^{-2}$ to over 25,834 cd m$^{-2}$, as shown in Supplementary Fig. 22d. The (PnP-QLED)$_{30}$ outputs very

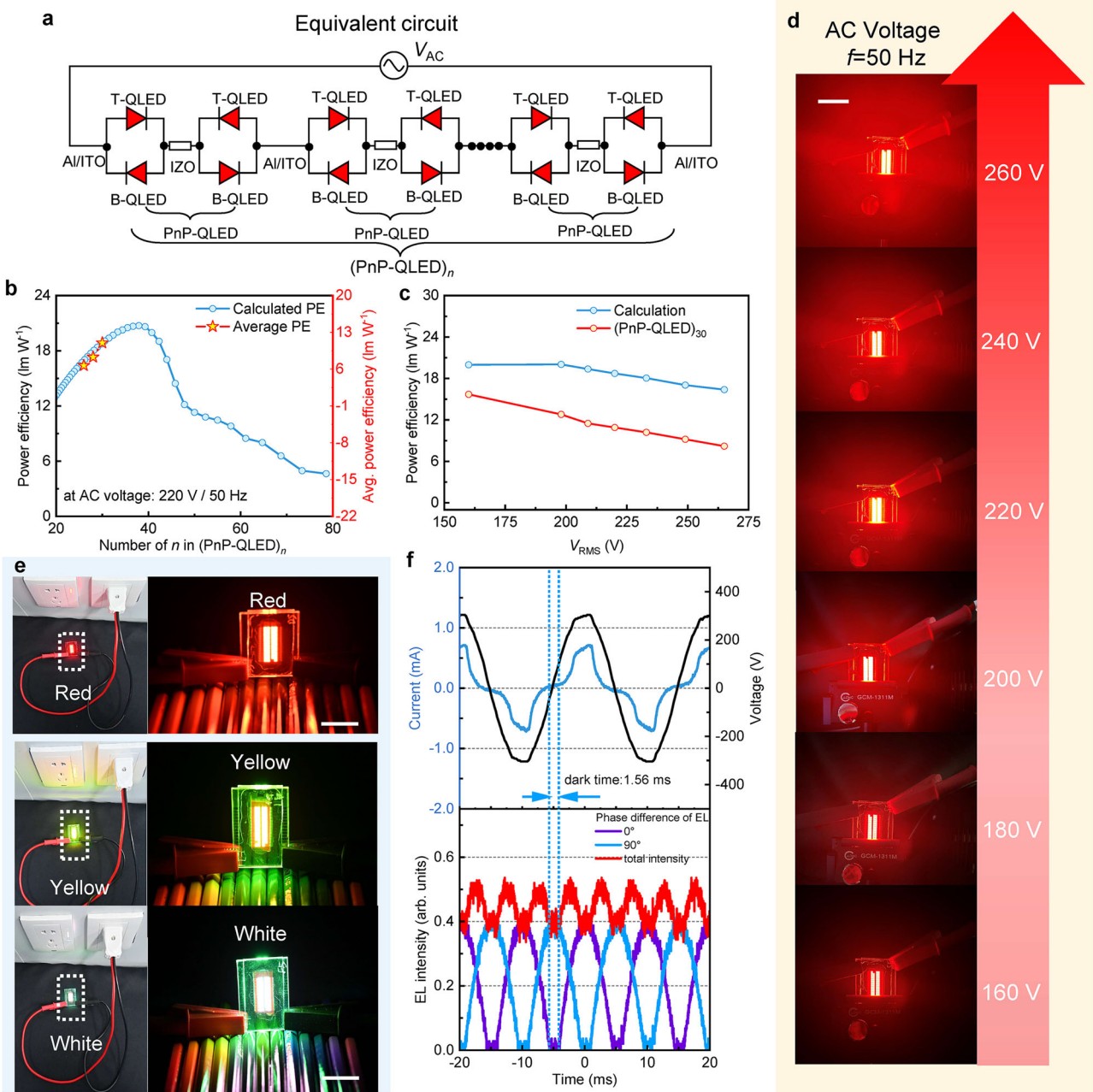

**Fig. 4 | 220 V/50 Hz directly driven (PnP-QLED)$_n$. a** Schematic circuit diagram of (PnP-QLED)$_n$ under AC driving. B-QLED/T-QLED represents the bottom/top QLED in single tandem QLED. PnP-QLED represents plug-and-play QLED. **b** The power efficiency (PE) of (PnP-QLED)$_n$ as a function of n under the driving of 220 V/50 Hz AC electricity (the blue circle/red star represents the calculated/testing PE of (PnP-QLED)$_n$). **c** The PE of (PnP-QLED)$_{30}$ as a function of driving voltage (the blue/red circle represents the calculated/testing PE of (PnP-QLED)$_{30}$). **d** The photos of (PnP-QLED)$_{30}$ driven by different AC voltage levels (also demonstrated in Supplementary Movie 3). Scale bars, 25 mm. **e** The photos of red (n = 30), yellow (n = 28), and white (n = 26) (PnP-QLED)$_n$ directly driven by 220 V/50 Hz household AC power supply (also demonstrated in Supplementary Movie 4). Scale bars, 20 mm. **f** The 220 V/ 50 Hz household AC voltage and the time-resolved electroluminescence (TrEL) of (PnP-QLED)$_{30}$ directly driven by 220 V/50 Hz household AC power supply with 0° (purple curve) and 90° (blue curve) phase, respectively (the red curve represents the total EL intensity of device under the combined driving of AC power supply with 0° and 90°).

uniform light emission and gradually enhanced brightness when the AC voltage is increased, as demonstrated in Fig. 4d and Supplementary Movie 3. The AC voltage can be simply tuned by changing the number of coils in the AC transformer (Supplementary Fig. 26), thereby providing an easy and cost-effective way for tuning the brightness of (PnP-QLED)$_n$. This approach not only reduces the AC-DC circuit but also eliminates the complex current regulation circuit that is used to tune the brightness of DC LEDs. As a demonstration, we show that our developed red (n = 30), yellow (n = 28), and white (n = 26) (PnP-QLED)$_n$ can be directly

plugged into a household 220 V/50 Hz power supply without needing any accessories, as shown in Fig. 4e and Supplementary Movie 4.

To investigate whether there is a luminance flickering that is commonly observed in AC-driven light sources such as incandescent bulbs or fluorescent lamps, the time-resolved electroluminescence (TrEL) of (PnP-QLED)$_{30}$ under household AC electricity driving was measured. As shown in Fig. 4f, the driving voltage varies in a sinusoidal manner, and at low driving voltage, the device cannot be switched on and therefore there is no light output. For example, for the

(PnP-QLED)$_{30}$ driven by a 220 V household power supply, there is 1.56 ms in an AC cycle (20 ms) that the device is in a dark state. Because the light output during an AC cycle is not continuous and steady, the human eye perceives luminance flickering. The luminance flickering can be mitigated by improving the continuity of light output. By reducing the value of $n$, the turn-on voltage of (PnP-QLED)$_n$ can be reduced, and thus the duration of the dark period can be decreased. For example, when the $n$ decreases from 30 to 16, the dark period of (PnP-QLED)$_n$ effectively decreases from 1.56 ms to 0.38 ms (Supplementary Fig. 27a), thereby leading to the suppression of luminance flickering. To achieve a more stable light output and further reduce luminance flickering, we connected two (PnP-QLED)$_{30}$ devices in parallel and modified the phase of the AC electricity for one (PnP-QLED)$_{30}$ device, as schematically illustrated in Supplementary Fig. 27b. The modification of the phase of the AC electricity can simply be achieved by using a capacitive delayer. By delaying 5 ms, the phase of the AC is altered by 90°. Under household AC electricity driving, the first (PnP-QLED)$_{30}$ is rapidly powered on, while the second (PnP-QLED)$_{30}$ is powered on 5 ms later. Due to the 90° phase difference of the AC driving, the EL of both (PnP-QLED)$_{30}$ is different at a given time. For example, when the first (PnP-QLED)$_{30}$ exhibit the maximum EL, the second (PnP-QLED)$_{30}$ exhibit the minimum (dark) EL, as shown in Fig. 4f. Therefore, the total EL from both (PnP-QLED)$_{30}$ is relatively stable and continuous, thus significantly reducing the luminance flickering.

## Discussion

In summary, we have developed an innovative tandem QLED that can be directly driven by household AC electricity. By using the IZO as an intermediate electrode to vertically connect the B-QLED and T-QLED, the resulting tandem QLED can be operated under DC or AC mode. Under DC mode, the tandem QLED exhibits a high EQE of 42.16%. Under AC mode, the B-QLED and T-QLED are alternately turned on with an EQE of 20.09% and 21.51%, respectively. Compared to conventional AC-EL devices, our AC-QLEDs exhibit continuous light emission throughout the entire driving cycle. In addition, they can be operated over a wide frequency range (several Hz-MHz) while maintaining high performance similar to that of S-QLEDs. Moreover, they demonstrate an extended $T_{95}$ at 10,000 cd m$^{-2}$ lifetime of 216 h, which is two times longer than that of S-QLEDs. With the developed AC-QLED as a basic building block, (PnP-QLED)$_n$ with $2n$ AC-QLEDs connected in series can be constructed. By varying the number of AC-QLEDs, the operational voltage of (PnP-QLED)$_n$ can be tuned to match arbitrary AC voltage levels. We demonstrate that our developed (PnP-QLED)$_n$ can be directly plugged into a household 220 V/50 Hz power socket. By varying the AC voltage, the brightness can be tuned from 1000 cd m$^{-2}$ to over 25,834 cd m$^{-2}$. The developed (PnP-QLED)$_n$ can be directly driven by arbitrary AC voltage and frequency without the need for complex back-end driver circuitry and thus could enable the production of low-cost, compact, efficient, and stable solid-state light sources for various applications.

## Methods

### Materials

All materials used in this study were commercially available. ITO glass with a sheet resistance of 20 Ω sq$^{-1}$ and a thickness of 80 nm was obtained from Wuhu Jinghui Electronic Technology Co., Ltd., while an IZO target consisting of 90 wt.% In$_2$O$_3$ and 10 wt.% ZnO was acquired from Hebei Gaocheng New Materials Technology Co., Ltd. Nanoparticles of ZnMgO (ZMO) were obtained from Guangdong Poly OptoElectronics Co., Ltd. The chemical 4,4′-Bis(9-carbazolyl)−1,1′-biphenyl (CBP) was acquired from Luminescence Technology Corp. Molybdenum trioxide (MoO$_3$) and octane were procured from Aladdin Industrial Corp. Absolute ethanol was sourced from Shanghai Ling-Feng Chemical Reagent Co., Ltd. Colloidal quantum dots in red

(CdZnSe/ZnS/oleic acid, core and shell ≈12.2 nm), green (CdZnSeS/ZnS/oleic acid, core and shell ≈11.6 nm), and blue (CdZnSeS/ZnS/oleic acid, core and shell ≈10.5 nm) varieties were sourced from Suzhou Xingshuo Nanotech Co., Ltd.

### Device fabrication

Inverted QLEDs: glass/ITO (80 nm)/ZMO ($x$ nm)/QDs (-15 nm)/CBP (50 nm)/MoO$_3$ (10 nm)/Al (100 nm), where the thickness of ZMO layer ranged from 40 nm to 150 nm.

Tandem QLEDs: glass/ITO ($x$ nm)/ZMO_B ($y$ nm)/QDs (-15 nm)/CBP (50 nm)/MoO$_3$ (10 nm)/IZO ($z$ nm)/ZMO_T ($k$ nm)/QDs (-15 nm)/CBP (50 nm)/MoO$_3$ (10 nm)/Al (100 nm) were fabricated. Abbreviation: ZMO_B = bottom ZnMgO, ZMO_T = top ZnMgO. The thicknesses of bottom ITO, ZMO_B, intermediate IZO, and ZMO_T were optimized to maximize the light outcoupling efficiency. By simulations and experiments, the optimal thicknesses of bottom ITO, ZMO_B, intermediate IZO, and ZMO_T are fixed as 80 nm, 110 nm, 25 nm, and 90 nm, respectively.

For the inverted QLEDs, the ITO glass substrates were cleaned with detergent and deionized water in an ultrasonic bath for 30 min and then dried in an oven at 60 °C for 30 min. After cleaning, the ZMO solution (40 mg mL$^{-1}$ in ethanol) and the QDs solution for red (R), green (G), blue (B), or yellow (Y) emission layers were layer-by-layer deposit by statically spin-casting at 3000 r.p.m. for 45 s, as shown in Supplementary Fig. 10a. The ZMO layer and QDs layer were then baked at 100 °C for 10 min and 5 min, respectively, to form the electron transport layer (ETL) and the respective emission layers (EML). For R-, G-, and B- QDs, the solutions were 15 mg mL$^{-1}$ in octane. For the Y-QDs emission layer, a mixed QDs solution was prepared by combining R-QDs (15 mg mL$^{-1}$ in octane) and G-QDs (15 mg mL$^{-1}$ in octane) with a volume ratio of 1:10. Subsequently, under a base pressure of $4 \times 10^{-4}$ Pa, the hole transport layer (HTL) of CBP and the hole injection layer (HIL) of MoO$_3$ were sequentially deposited onto the QDs layer at rates of 1.5 Å s$^{-1}$ and 0.2 Å s$^{-1}$, respectively. Then, Al electrodes (100 nm) were deposited using the same method at a rate of 5 Å s$^{-1}$. The device area defined by the overlapping of the ITO and Al electrodes is 1 mm$^2$. Finally, the QLEDs were encapsulated with UV-resin and cover glass.

For tandem QLEDs, the fabrication processes for the first part, glass/ITO/ZMO_B/QD/CBP/MoO$_3$, mirrored those of the inverted QLED. Following the MoO$_3$ deposition, the samples were transferred to a magnetron sputtering system to deposit the IZO electrode at varying thicknesses under a working pressure of 0.45 Pa, a power of 50 W, and an Ar flow of 20 sccm. Subsequently, the ZMO solution (40 mg mL$^{-1}$ in ethanol) and the QDs solution were layer-by-layer deposited by dynamically spin-casting at 3000 r.p.m. for 45 s, as shown in Supplementary Fig. 10b. The ZMO layer and QD layer were then baked at 60 °C for 20 min and 10 min, respectively, in a nitrogen-filled glovebox to form the ETL (ZMO_T) and EML. Following this, the CBP, MoO$_3$, and Al were thermally deposited under the same conditions as the inverted QLEDs. The device area defined by the overlapping of the ITO, IZO, and Al electrodes is 1 mm$^2$. Finally, the QLEDs were encapsulated with UV-resin and cover glass. Supplementary Fig. 11 illustrates the fabrication processes developed to achieve tandem QLEDs without damage.

### Characterizations

The thicknesses of ITO, IZO, ZMO, and QD film were determined utilizing a Bruker DektakXT stylus profiler. In situ monitoring of the evaporation rates and thicknesses of CBP, MoO$_3$, and Al electrodes was conducted using a quartz crystal microbalance. The EL spectra of QLEDs were recorded in the standard direction using a fiber-optic spectrometer (Ocean Optics USB 2000). The $J$-$V$-$L$, EQE, and PE characteristics of QLEDs were obtained using a home-built system comprising a dual-channel Keithley 2614B programmable source meter and a PIN-25D calibrated silicon photodiode (PD). To measure the performance of (PnP-QLED)$_n$, the (PnP-QLED)$_n$ were driven by AC electricity

at 50 Hz using an isolation transformer. The AC voltage dropped across the (PnP-QLED)$_n$ and the current flowing through the (PnP-QLED)$_n$ were recorded by a dual-channel oscilloscope (Tektronix, TBS1102). The luminous flux generated by the (PnP-QLED)$_n$ was measured by a PIN-25D calibrated silicon photodiode. The schematic measurement setup is shown in Supplementary Fig. 21. The average of PE, EQE, and CE of the (PnP-QLED)$_n$ were calculated by using the measured $I_{avg}$, $P_{avg}$, lumens, and spectra of the devices as the inputs.

## Data availability

The data that support the findings of this study are available from the corresponding author upon request.

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

## Acknowledgements

This work was supported by the National Natural Science Foundation of China, No. 62174075 (S.C.), Shenzhen Science and Technology Program, Nos. JCYJ20210324105400002 (S.C.), JCYJ20220530113809022 (S.C.) and JCYJ20230807093604009 (S.C.).

## Author contributions

S.C. conceived the idea, supervised the work, and wrote the majority of the final manuscript. J.W. conducted the experiments, collected the data, drew the figures, and wrote the original draft. C.Y. discussed the optical simulation results. All authors discussed the results and reviewed the manuscript.

## Competing interests

The authors declare no competing interests.
