## [Peer Review File · Nature Communications]

Household alternating current electricity plug-and-play
quantum-dot light-emitting diodesREVIEWER COMMENTS

Reviewer #1 (Remarks to the Author):

The authors presented tandem PnP QLEDs driven by household AC voltage. IZO was used as an intermediate layer; the simple tandem device (S-OLED) driven under DC voltage exhibited an impressive performance of more than 40% EQE. The PnP device is quite clever; the intermediate IZO layer does not need to be connected to the external electrode directly, which can be an advantage in actual device fabrication because of no need to add contact holes. The lifetime was extended owing to the tandem effect and the negative bias effect. Frankly speaking, I am skeptical about the main goal of this study; the function and performance of external electronic circuits may not be replaced by the clever QLED device structure. The use of a serially connected 38 devices is not practical. However, it may deserve its publication considering the intriguing idea and high performance. Please, consider the following.

1. Actual household AC voltage is a sinusoidal wave rather than a rectangular wave. Therefore, the issues regarding luminance flickering and low efficiency at low voltage can partially occur. It should be discussed.
2. Normally, CRI, CCT, and spectrum are required for lighting applications. Adding any of these data will be helpful.
3. Asymmetric losses for negative and positive biases may need further discussion.

Reviewer #2 (Remarks to the Author):

In this study, the authors reported a novel tandem QLED that can be directly powered by household AC electricity by using an intermediate electrode to vertically connect the B-QLED and T-QLED. The tandem QLED can be operated with a high external quantum efficiency. Furthermore, they also demonstrated a household electricity plug-and-play (PnP) light source, which can be directly driven by 220/110 V AC electricity without the need for complicated back-end circuits, by connecting multiple tandem QLEDs in series.

1. First, a comparison of the work with the previously reported AC EL device with insulating layers is not fair since the devices with insulators didn't work under DC. The authors should carefully compare the performance of their device with ones capable of operating under DC as well as AC. Particularly, the AC-driven light-emitting diodes featuring tandem structures containing an intermediate electrode have been studied. I notice that several works have been done by the authors. Careful comparisons should be made to claim the novelty of the current work. For example, the authors highlight the advantages of the tandem QLED over other AC-EL devices, noting its (1) continuous light emission, (2) high performance including a longer lifetime, and (3) low-frequency operation. All EL devices with this tandem structure with intermediate electrodes, however, possess continuous light emission. More importantly, since they consist of connected regular and inverted structure diodes, they are all capable of operating at low frequencies. Although most of the works were focused on color tunability using AC electricity, some studies demonstrate sufficiently high performance. PE of 36.8 lm/W @ 1000 cd/m² was reported in the paper of Light-Sci. Appl. 4, e247, 2015. The EQE of 26.02 % and high brightness of 107,000 cd/m² were shown in the authors' previous work (Nat. Comm. 11, 2826, 2020.) I also noticed that the authors have successfully demonstrated tandem devices by combining red and green QLEDs that reliably operated for over 25,000 hours with commercially available blue OLEDs, focusing on the aspect of lifetime. (Nat. Comm. 11, 2826, 2020., Nanoscale 13, 16781, 2021.) In addition, the details of the tandem QLED under AC mode (Figure 2) should be shown in Table 1 with the results

of the device under DC mode to see the difference more clearly between the two modes. For example, what about the PE of the device in AC mode under 10,000 cd/m² rather than 1000 cd/m², as shown in Table 1?

2. The novelty I can see is that well-performing tandem AC EIs which have been previously demonstrated in several groups were connected in series, capable of operating under household AC electricity. I feel doubtful if such technical modification in electrical connection is sufficiently novel. If this connection technique is critical for household AC operation, a more systematic study should be performed to ensure that the method present in the work indeed outruns the AC-DC converter technology. For instance, in the section concerning the direct household AC electricity-driven PnP-QLED on page 13, the author highlighted that connecting a maximum of 30 PnP-QLEDs yielded optimal results, aligning with the calculated expectations. The lifetime of the connected devices should be provided.

3. In Figure 4 and Figure S17, there seems to be a consistent, slight depression of Power Efficiency, Current Efficiency, and EQE around 175V in ((PnP-QLED)) _30. For ((PnP-QLED)) _26, ((PnP-QLED)) _28 case, there was a slight increase in PE, CE, and EQE at around 175V, whereas for ((PnP-QLED)) _30, it showed a decrease in such regions.

However, there seems to be insufficient explanation regarding this matter. It is suggested that the authors clarify whether this is due to experimental error or consistent phenomena and provide an explanation regarding this matter.

4. On page 2, the authors claim that removing the DC drivers and AC-DC converters results in energy losses and increased power consumption in the conversion process. Although we can agree on the fact that removing these complicated back-end circuits is beneficial in securing energy efficiency in general, given that there is noticeable energy loss in negative driving cycles, it can be suggested to add data regarding the comparison of efficiency between conventional household electronics (with circuitry) and this proposed work (without circuitry).

5. In Figure 2 and Figure S10, there is a considerable difference between losses (EQE, CE, PE) in negative driving conditions compared to positive driving conditions. Although the article mainly focuses on the high current range where losses can be neglected, it seems like more explanation can be suggested of why there is a deviation of loss between positive/negative driving conditions.

6. The authors show optical simulation results of the tandem QLED in Fig. 1c, Fig. S4, and Fig. S5 to optimize the thickness of the layers of the device. It would be beneficial to have a detailed description of the simulation conditions and methods, along with an extended discussion of the results. On page 5 of the main text, the author claims that the size of one cavity is impacted by the size of another cavity. As a result, the tandem device was conceived as two Fabry-Pérot resonant cavities stacked vertically. However, the author presented simulated data without accompanying explanations. Hence, a comprehensive clarification of the simulation principles and results is necessary. Furthermore, since these results are purely from simulations, empirical data would be provided to validate these findings.

7. In Figure 1e, the luminance data for B-QLED and T-QLED ceases beyond 8V, whereas S-QLED continues to persist beyond 16V. Is the absence of data beyond 8V indicative of device damage, or is there another underlying reason? If that's the case, at what voltage was the device's lifetime tested and driven in Figure 2d?

8. In Figure S17, the legends in Figures A, b, c are labeled inconsistently. Figure S17 a, c's legend is labeled "the ((PnP-QLED)) _26 ", where for Figure S17 b's legend is labeled, " ((PnP-QLED)) _26 ". To prevent confusion, correction is suggested.

9. Some scale bars are missing in the photographs of Figures.

Reviewer #3 (Remarks to the Author):

The authors reported their work on household AC electricity plug-and-play quantum-dot light-emitting diodes. They made high-performance tandem QLEDs, which can be operated at both negative and positive cycles of the AC voltage, by using a transparent and conductive IZO as an intermediate electrode resulting in high EQEs of 22.99%. The idea is good and the experimental data are reliable. However, there are a few issues to be addressed before it could be published in this journal.

1. Although it would be cost-effective that their tandem QLEDs can be directly driven by 220/110 V AC electricity without the need for complicated back-end circuits, the AC-QLEDs, similar to some other AC driven lighting sources, cannot provide with real continuous light emission with steady light output. Actually, in-house lighting needs a steady light output for eye health. How to provide with steady light output is a big challenge to this AC-QLEDs. The authors should find a way to resolve this problem. What is the dark interval time when the voltage was changed from the positive to the negative?

2. The authors mentioned regarding Movie 3 that “the flicker is due to the mismatch of the QLED lighting frequency and the frame captured frequency of the camera. The emission is quite stable observed by eyes”. However, it needs measurement data to prove the statement. Therefore, it is suggested the authors should show the relationship between light output vs. time under $V(\text{RMS}) = 220 \text{ V}$ with a chart similar to that in Figures S19 and S20.

Other than the aforementioned major issue, there are a few format typos:

Fig S2a: The $V\text{-AC}$ on the top should be put in one line. Fig. S2c: Since Electrodes A and B contact to the same charge generation layer, the energy levels of both electrodes relating to the HOMO of the charge generating layer should be the same.

Fig. S4: In the annotation, “b, Simulated outcoupling efficiency of b, B-QED,...ITO”: It might be better to put the chart numbers in the front of the sentence. How about “b, c, and d, Simulated outcoupling efficiency of B-QLED, T-QLED, and S-QLED, respectively, as a function of ...ITO”? “B-QED” should be “B-QLED”.

Fig. S4: In the annotation: “Power fraction of each mode as a fraction of the ITO thickness in red e, B-QLED...”, What is the meaning of “in red e”? Red color in the chart is only the air mode of the power fraction.

Fig. S5: “80 nm, 20 nm, 32%” is good for the B-QLED, “25 nm, 110 nm, 32%” is good for the T-QLED. How do you conclude that “30 nm, 90 nm” is good for the S-QLED?

Fig. S6: In order to understand the dynamic spin-coating, it is suggested to show the static spin-coating condition in this work.

Fig. S7: On the bottom-right, “B-QLED” should be “T-QLED”.

Fig. S8: 1) The thickness of D1 is also critical to the outcoupling of the light. However, there is no any discussion on D1. 2) If caption “ZMO thickness” is used, “Current Density” should be changed as “Current density”. This “capital initial letter” unconsistence can also be seen

in the other figures, such as Figs. S9, S10, S17.

Response to Reviewer #1

General comment: *The authors presented tandem PnP QLEDs driven by household AC voltage. IZO was used as an intermediate layer; the simple tandem device (S-OLED) driven under DC voltage exhibited an impressive performance of more than 40% EQE. The PnP device is quite clever; the intermediate IZO layer does not need to be connected to the external electrode directly, which can be an advantage in actual device fabrication because of no need to add contact holes. The lifetime was extended owing to the tandem effect and the negative bias effect. Frankly speaking, I am skeptical about the main goal of this study; the function and performance of external electronic circuits may not be replaced by the clever QLED device structure. The use of a serially connected 38 devices is not practical. However, it may deserve its publication considering the intriguing idea and high performance. Please, consider the following.*

Our response: Thank you for your efforts in reviewing this manuscript. We sincerely appreciate your positive comments such as “clever PnP device structure, impressive performance, extended lifetime, an advantage in actual device fabrication”, *etc.* Also, your constructive suggestions are very helpful, which greatly help us to improve the quality of this paper.

For your concern about the goal of this study, we would like to further clarify the advantages of the proposed (PnP-QLED)_n, as summarized below.

- (1) The (PnP-QLED)_n can be directly driven by household AC voltage without the need for complex back-end driver circuitry (including AC-DC adapter and DC-DC converter which account for approximately 17% of the entire LED lamp cost)¹, and thus **reduces the cost of the QLED lamps.**
- (2) The AC driven (PnP-QLED)_n exhibit a **2-fold longer lifetime** compared to the DC-QLED while maintaining the same efficiency, as you have pointed out.
- (3) The elimination of driver circuitry not only reduces the cost, but also **avoids the energy losses** (10~20%) during the AC-DC conversion and the DC-DC regulation processes²⁻⁶.

For your concern about the feasibility, the (PnP-QLED)_n was realized by connecting multiple tandem QLEDs in series, which can be easily fabricated by using the shadow masks to define the electrodes. The fabrication processes are almost the same as those of the conventional QLEDs, so it is practical and feasible.

References:

1. Lee, K., Nubbe, V., Rego, B., Hansen, M. & Pattison, P. 2020 LED manufacturing

supply chain. (2021).

2. Chiu, H.-J., Lo Yu-Kang, Chen, J.-T., Cheng, S.-J., Lin, C.-Y. & Mou, S.-C. A high-efficiency dimmable LED driver for low-power lighting applications. *IEEE Trans. Ind. Electron.* 57, 735-743 (2010).
3. Yu, W., Lai, J.-S., Ma, H. & Zheng, C. High-efficiency DC-DC converter with twin bus for dimmable LED lighting. *IEEE Trans. Power Electron.* 26, 2095-2100 (2011).
4. Pollock, A., Pollock, H. & Pollock, C. High efficiency LED power supply. *IEEE J. Emerg. Sel. Topics Power Electron.* 3, 617-623 (2015).
5. Zhang, F., Ni, J. & Yu, Y. High power factor AC-DC LED driver with film capacitors. *IEEE Trans. Power Electron.* 28, 4831-4840 (2013).
6. Malcovati, P., Belloni, M., Gozzini, F., Bazzani, C. & Baschiroto, A. A 0.18- μm CMOS, 91%-efficiency, 2-A scalable buck-boost DC-DC converter for LED drivers. *IEEE Trans. Power Electron.* 29, 5392-5398 (2014).

Comment #1: Actual household AC voltage is a sinusoidal wave rather than a rectangular wave. Therefore, the issues regarding luminance flickering and low efficiency at low voltage can partially occur. It should be discussed.

Response #1: Thanks for your insightful comments. We fully agree that luminance flickering does occur when driving the devices with a sinusoidal AC source. This is because the driving voltage varies in a sinusoidal manner and if the driving voltage is low, the device cannot be switched on and therefore there is no light output. Because the light output during an AC cycle is not continuous and steady, the human eye perceives luminance flickering. Actually, **luminance flickering is a common issue that can be observed in all AC light sources** such as the incandescent bulbs or fluorescent lamps. For almost all light sources, there is usually a trade-off between lamp cost and luminance flickering.

For our (PnP-QLED)_n, the luminance flickering can be mitigated by improving the continuity of the light output. To this end, we propose a method involving the parallel connection of two (PnP-QLED)_n devices, with a modification of the AC electricity phase for one (PnP-QLED)_n device, as illustrated in Figure R1a (Supplementary Figure S24b in the revised manuscript). The modification of the phase of the AC electricity can simply be achieved by using a capacitive delayer. By delaying 5 ms, the phase of the AC is altered by 90°. Under household AC electricity driving, the first (PnP-QLED)_n is rapidly powered on, while the second (PnP-QLED)_n is powered on 5 ms later. When the first (PnP-QLED)_n exhibit the maximum EL, the second (PnP-QLED)_n exhibit the minimum (dark) EL, as shown in Figure R1b (Fig.

4f in the revised manuscript). Therefore, the total EL from both (PnP-QLED)_n is relatively stable and continuous, thus significantly reducing the luminance flickering.

To clarify this point, a few sentences had been added to the revised manuscript, as follows. (Page 15-16, line 353-377 in the revised manuscript): “To investigate whether there is a luminance flickering that is commonly observed in AC driven light sources such as incandescent bulbs or fluorescent lamps, the time-resolved electroluminescence (TrEL) of (PnP-QLED)₃₀ under household AC electricity driving was measured. As shown in Fig. 4f, the driving voltage varies in a sinusoidal manner, and at low driving voltage, the device cannot be switched on and therefore there is no light output. For example, for the (PnP-QLED)₃₀ driven by a 220 V household power supply, there are 1.56 ms in an AC cycle (20 ms) that the device is in a dark state. Because the light output during an AC cycle is not continuous and steady, the human eye perceives luminance flickering. The luminance flickering can be mitigated by improving the continuity of light output. By reducing the value of n, the turn-on voltage of (PnP-QLED)_n can be reduced, and thus the duration of the dark period can be decreased. For example, when the n decreases from 30 to 16, the dark period of (PnP-QLED)_n effectively decreases from 1.56 to 0.38 ms (Supplementary Fig. S24a), thereby leading to the suppression of luminance flickering. To achieve a more stable light output and further reduce luminance flickering, we connected two (PnP-QLED)₃₀ devices in parallel and modified the phase of the AC electricity for one (PnP-QLED)₃₀ device, as schematically illustrated in Supplementary Fig. S24b. The modification of the phase of the AC electricity can simply be achieved by using a capacitive delayer. By delaying 5 ms, the phase of the AC is altered by 90°. Under household AC electricity driving, the first (PnP-QLED)₃₀ is rapidly powered on, while the second (PnP-QLED)₃₀ is powered on 5 ms later. Due to the 90° phase difference of the AC driving, the EL of both (PnP-QLED)₃₀ is different at a given time. For example, when the first (PnP-QLED)₃₀ exhibit the maximum EL, the second (PnP-QLED)₃₀ exhibit the minimum (dark) EL, as shown in Fig. 4f. Therefore, the total EL from both (PnP-QLED)₃₀ is relatively stable and continuous, thus significantly reducing the luminance flickering.”

Figure R1. **a**, Schematic circuit diagram of two (PnP-QLED)_n under 220 V/50 Hz AC driving (reprinted from Supplementary Figure S24b of the revised manuscript). **b**, The TrEL of (PnP-QLED)₃₀ directly driven by 220 V/ 50 Hz household AC power supply with different phase difference. (reprinted from Fig. 4f of the revised manuscript)

As for your concern on the low efficiency at low voltage, we also agree with you. However, because at low voltage, the device cannot be turned-on, the current is very low (Figure R1b) and the device therefore consumes almost no energy. After the device is turned-on, the efficiency rapidly climbs to its peak value. Therefore, the influence of the low efficiency at low voltage does not significantly reduce the overall efficiency of the (PnP-QLED)_n over the entire AC cycle.

Comment 2: Normally, CRI, CCT, and spectrum are required for lighting applications. Adding any of these data will be helpful.

Response #2: Thanks for your helpful suggestion. In Figure R2 (Supplementary Figure S9a in the revised manuscript), we have provided the emission spectrum of the red QLED. Because we mainly used the red QDs to demonstrate the concept of PnP-QLED, the CRI, CCT are not applicable to our red devices. In the future, we will fabricate the white PnP-QLED, and shall calculate its CRI and CCT.

Figure R2 The normalized EL spectra of B-QLED (red), T-QLED (green), and S-QLED (blue). (reprinted from Supplementary Figure S9a of the revised manuscript)

Comment 3: *Asymmetric losses for negative and positive biases may need further discussion*

Response #3: Thanks for your suggestion. As shown in Figure R3 (Fig. 2b in the revised manuscript), during the negative driving cycle, the reverse current in AC-QLED is much higher compared to that during the positive driving cycle. Consequently, at low currents or low luminance levels, AC-QLED driven in the negative cycle has higher energy losses.

To clarify this point, a few sentences had been added to the revised manuscript, as follows (page 9-10, line 218-227 in the revised manuscript): “At low current or low brightness levels, the reverse current cannot be neglected, as it greatly affects the efficiency losses of the AC-QLED. As shown in Fig. 2c, it can be observed that the efficiency losses of AC-QLED under positive and negative driving cycles (green shaded areas) are not equal. Due to the rougher surface induced by the thick bottom layers, the reverse current J_{R_T} (from the T-QLED) during the negative driving cycles is much higher than the reverse current J_{R_B} (from the B-QLED) during the positive driving cycles, as illustrated in Fig. 2b. Consequently, under negative driving cycles, the energy losses in the AC-QLED are higher compared to those under positive driving cycles. However, for high currents or high brightness situations, because the forward current is significantly higher than the reverse current, the impact of reverse current on the efficiency of AC-QLED becomes negligible.”

Figure R3 J - V characteristics of T-QLED, B-QLED, and AC-QLED in positive driving cycles (top) and negative driving cycles (bottom). (reprinted from Fig. 2b of the revised manuscript)

To address the inconsistency in energy loss of AC-QLED under positive and negative driving cycles at low currents and low brightness levels, we subsequently developed the PnP-QLED as the fundamental unit for household AC driving device. The schematic circuit diagram of the PnP-QLED is shown in Figure R4a (Fig. 3b in the revised manuscript). In PnP-QLED, regardless of the positive or negative driving cycles, the reverse current in the PnP-QLED is always determined by both J_{R_B} and J_{R_T} . As depicted in Figure R4b-e, the performance of PnP-QLED remains the same during both positive and negative driving cycles.

Figure R4 a, schematic circuit diagrams and photos of a PnP-QLED under the positive (top) and the negative (bottom) driving cycles. (reprinted from Fig 3b of the revised manuscript) **b-e**, The J - V - L , CE - J , EQE - J and PE - J characteristics of PnP-QLED under positive and negative driving cycles, respectively.

We hope our responses/revisions satisfactorily address all your concerns. Once again, we thank you for your constructive and helpful suggestions!

Response to Reviewer #2

General comment: *In this study, the authors reported a novel tandem QLED that can be directly powered by household AC electricity by using an intermediate electrode to vertically connect the B-QLED and T-QLED. The tandem QLED can be operated with a high external quantum efficiency. Furthermore, they also demonstrated a household electricity plug-and-play (PnP) light source, which can be directly driven by 220/110 V AC electricity without the need for complicated back-end circuits, by connecting multiple tandem QLEDs in series.*

Our response: Thanks for your efforts in reviewing this manuscript. We sincerely appreciate your comments that certainly help to improve the quality of this paper.

Comment #1: *First, a comparison of the work with the previously reported AC EL device with insulating layers is not fair since the devices with insulators didn't work under DC. The authors should carefully compare the performance of their device with ones capable of operating under DC as well as AC. Particularly, the AC-driven light-emitting diodes featuring tandem structures containing an intermediate electrode have been studied. I notice that several works have been done by the authors. Careful comparisons should be made to claim the novelty of the current work. For example, the authors highlight the advantages of the tandem QLED over other AC-EL devices, noting its (1) continuous light emission, (2) high performance including a longer lifetime, and (3) low-frequency operation. All EL devices with this tandem structure with intermediate electrodes, however, possess continuous light emission. More importantly, since they consist of connected regular and inverted structure diodes, they are all capable of operating at low frequencies. Although most of the works were focused on color tunability using AC electricity, some studies demonstrate sufficiently high performance. PE of 36.8 lm/W @ 1000 cd/m² was reported in the paper of Light-Sci. Appl. 4, e247, 2015. The EQE of 26.02 % and high brightness of 107,000 cd/m² were shown in the authors' previous work (Nat. Comm. 11, 2826, 2020.) I also noticed that the authors have successfully demonstrated tandem devices by combining red and green QLEDs that reliably operated for over 25,000 hours with commercially available blue OLEDs, focusing on the aspect of lifetime. (Nat. Comm. 11, 2826, 2020., Nanoscale 13, 16781, 2021.) In addition, the details of the tandem QLED under AC mode (Figure 2) should be shown in Table 1 with the results of the device under DC mode to see the difference more clearly between the two modes. For example, what about the PE of the device in AC mode under 10,000 cd/m² rather than 1000 cd/m², as shown in Table 1?*

Response #1: Thanks for your insightful comments and helpful suggestions. We address each of your comments point-by-point below.

- (1) We agree that it is fairer to compare the performance of our AC-QLEDs with ones capable of operating under DC as well as AC. Actually, Table R1 (Supplementary Table S1 in the revised manuscript) already outlined this comparison. We noticed that some white OLEDs (*Light-Sci. Appl.* 4, e247, 2015) do exhibit impressive power efficiency of 36.8 lm/W @ 1000 cd/m² and some tandem organic-quantum dot hybrid white LEDs do exhibit high EQE of 26.02% and high brightness of 107,000 cd/m² (as we reported in *Nat. Comm.* 11, 2826, 2020). Because the emission spectra are different, it is fairer to compare the EQE of the devices. As shown in Table R1, our red AC-QLED demonstrate **a high EQE of ~20% @ 1000 cd/m²**, which is much higher than 15~16% reported in *Light-Sci. Appl.* 4, e247, 2015. Our tandem red QLED (equivalent to a PnP-QLED) shows **a high EQE of 37.05% @ 1000 cd/m²**, which is remarkable higher than 26.02% reported in *Nat. Comm.* 11, 2826, 2020. In terms of stability, as shown in Figure R5a (Fig. 2e in the revised manuscript), our AC-QLED outperforms DC-QLED **by a factor of 2. We can safely conclude that with our new device structure and improved fabrication process, the performance including both EQE and lifetime of our AC-QLEDs are much better than previously reported ones.**
- (2) We agree that the reported AC-ELs (capable of operating under DC as well as AC) also have the three advantages (continuous light emission, high performance and low-frequency operation) that we claimed. However, **no AC-EL devices have been demonstrated that can be directly driven by standard household electricity (110/220 V, 50/60 Hz). Our proposed (PnP-QLED)_n is the first EL device that can be directly plugged into a household 220 V/50 Hz power socket.** To clarify this point, a few sentences had been added to the revised manuscript, as follows (page 3-4, line 86-90 in the revised manuscript): “As summarized in Supplementary Fig. S3 and Supplementary Table S1, the reported AC-EL devices driven by high AC voltage exhibit lower performance (brightness and power efficiency) than that of the DC devices, and thus far, there is no AC-EL device that can be directly driven by 110/220 V at 50/60 Hz household electricity with high efficiency and long lifetime.”
- (3) Regarding your concern about the performance comparison of the tandem QLED operated between AC mode and DC mode, we actually already presented such a comparison in Fig. 2c and Supplementary Figure S10. When the tandem QLED operates in AC mode, both T-QLED and B-QLED are alternately turned-on, and therefore we compared the performance of the AC-QLED with those of T-QLED or B-QLED that operates in DC-mode. As shown in Figure R5b-c (Supplementary Figure S10 in the revised manuscript),

under AC mode, the efficiency of AC-QLED is almost equal to that of DC-QLED (T-QLED or B-QLED), especially in the high current range. At low current and low brightness level, the performance of AC-QLED is a bit lower than that of the DC-QLED, due to the presence of the reverse current. We have fully discussed the performance difference of the tandem QLED operated between AC mode and DC mode in the main text of the manuscript (page 9, line 218-239 in the revised manuscript).

- (4) Following your suggestion, we had included the PE data under 1000 cd/m² in Table 1.

Table R1 Performance comparison of AC-driven EL devices (reprinted from Supplementary Table S1 of the revised manuscript).

Type	Device architecture	EL (nm)	PE@1,000 cd/m ² (lm/W)	Max Luminance (cd/m ²)	Max V _{RMS} (V)	Optimal Frequency (Hz)	Response of EL in AC voltage cycle	Compatible with the household AC power lines	Ref	
	ITO/Au/Ag/Spiro-TTB:F6-TCNNQ/Spiro-TAD/O-EM/Bphen/Bphen:Cs/Au/Ag/Spiro-TTB:F6-TCNNQ/Spiro-TAD/O-EM/Bphen/Bphen:Cs/Au/Ag/ α -NPD	White	9.7	N/A	N/A	50	Full cycle	NO	4	
	ITO/Spiro-TTB:F6-TCNNQ/Spiro-TAD/ α -NPD/4P-NPD/Bphen/BAIq ₃ /Bphen:Cs/Au/Ag/Spiro-TTB:F6-TCNNQ/Spiro-TAD/TCTA:Ir(dhfp) ₂ (acac)/TPBi:Ir(dhfp) ₂ (acac)/BAIq ₃ /Bphen:Cs/Al	White	36.8 (Yellow: 15-16% EQE; Blue: 3.4% EQE)	N/A	N/A	50	Full cycle	NO	1	
Multi electrodes	ITO/PEDOT:PSS/TFB/QD/ZnO/ultra-thin Al/IZO/MoO ₃ /NPB/MADN:DSA-Ph/TPBi/LiF/Al	White	N/A	107,000	14.0	50	Full cycle	NO	2	
	ITO/PEDOT:PSS/TFB/QD/ZnO/ultra-thin Al/IZO/MoO ₃ /NPB/MADN:DSA-Ph/TPBi/LiF/Al	Red	N/A	18,750	8.0	50	Full cycle	NO	3	
	ITO/ZnO/QD/CB P/MoO ₃ /IZO/ZnO/QD/CB P/MoO ₃ /Al	AC-QLED	Red	23.16 (20.11% EQE)	15,700	6.0	50	Full cycle	NO	
		PnP-QLED	Red	21.44 (37.05% EQE)	32,265	12.0	50	Full cycle	NO	Our work
		(PnP-QLED) _n	Red	15.7 (770% EQE)	25,834	220	50	Full cycle	YES	

References:

- Fröbel, M., Schwab, T., Kliem, M., Hofmann, S., Leo, K. & Gather, M. C. Get it white: color-tunable AC/DC OLEDs. *Light Sci Appl* 4, e247-e247 (2015).

2. Zhang, H., Su, Q. & Chen, S. Quantum-dot and organic hybrid tandem light-emitting diodes with multi-functionality of full-color-tunability and white-light-emission. *Nat. Commun.* 11, 2826 (2020).
3. Zhang, H., Chen, L. & Chen, S. Quantum-dot and organic hybrid tandem light-emitting diodes with color-selecting intermediate electrodes for full-color displays. *Nanoscale* 13, 16781–16789 (2021).
4. Fries, F., Fröbel, M., Lenk, S. & Reineke, S. Transparent and color-tunable organic light-emitting diodes with highly balanced emission to both sides. *Org. Electron.* 41, 315–318 (2017).

Figure R5 AC-QLED. **a**, The lifetime curves of the S-QLED (red curve) and AC-QLED (blue curve) (reprinted from Fig. 2e of the revised manuscript). **b** and **c**, CE - J and PE - L characteristic curves of devices under positive or negative driving cycles, respectively. (reprinted from Supplementary Figure S10 of the revised manuscript) Because the reverse current is relatively small, the devices that are reversely biased do not incur appreciable energy losses to the AC-QLED, especially at high current density or high brightness.

Comment #2: *The novelty I can see is that well-performing tandem AC ELs which have been previously demonstrated in several groups were connected in series, capable of operating under household AC electricity. I feel doubtful if such technical modification in electrical connection is sufficiently novel. If this connection technique is critical for household AC operation, a more systematic study should be performed to ensure that the method present in the work indeed outruns the AC-DC converter technology. For instance, in the section concerning the direct household AC electricity-driven PnP-QLED on page 13, the author highlighted that connecting a maximum of 30 PnP-QLEDs yielded optimal results, aligning with the calculated expectations. The lifetime of the connected devices should be provided.*

Response #2: Thanks for your valuable comments. We address each of your comments point-by-point below.

- (1) Regarding your concern about the novelty, we agree that tandem AC-EL have been previously demonstrated by other groups or us; however, **no AC-EL devices have been demonstrated that can be directly driven by standard household electricity (110/220 V, 50/60 Hz). Our proposed (PnP-QLED)_n is the first EL device that can be directly plugged into a household 220 V/50 Hz power socket, while retains high efficiency and long lifetime.**
- (2) Regarding your concern about the advantages of our (PnP-QLED)_n over traditional DC-QLED, we would like to clarify that our method outruns the AC-DC technology in terms of:
 - i. The (PnP-QLED)_n can be directly driven by household AC voltage without the need for complex back-end driver circuitry (including AC-DC adapter and DC-DC converter which account for approximately 17% of the entire LED lamp cost ¹), and thus **reduces the cost of the QLED lamps.**
 - ii. The AC driven (PnP-QLED)_n exhibit a **2-fold longer lifetime** compared to the DC-QLED while maintaining the same efficiency, as shown in Figure R7a (Fig. 2d-e in the revised manuscript).
 - iii. The elimination of driver circuitry not only reduces the cost, but also **avoids the energy losses (10~20%)** ²⁻⁶ during the AC-DC conversion and the DC-DC regulation processes.

References:

1. Lee, K., Nubbe, V., Rego, B., Hansen, M. & Pattison, P. 2020 LED manufacturing supply chain. (2021).
2. Chiu, H.-J., Lo Yu-Kang, Chen, J.-T., Cheng, S.-J., Lin, C.-Y. & Mou, S.-C. A high-efficiency dimmable LED driver for low-power lighting applications. *IEEE Trans. Ind. Electron.* 57, 735-743 (2010).
3. Yu, W., Lai, J.-S., Ma, H. & Zheng, C. High-efficiency DC-DC converter with

twin bus for dimmable LED lighting. *IEEE Trans. Power Electron.* 26, 2095-2100 (2011).

4. Pollock, A., Pollock, H. & Pollock, C. High efficiency LED power supply. *IEEE J. Emerg. Sel. Topics Power Electron.* 3, 617-623 (2015).
5. Zhang, F., Ni, J. & Yu, Y. High power factor AC-DC LED driver with film capacitors. *IEEE Trans. Power Electron.* 28, 4831-4840 (2013).
6. Malcovati, P., Belloni, M., Gozzini, F., Bazzani, C. & Baschirotto, A. A 0.18- μm CMOS, 91%-efficiency, 2-A scalable buck-boost DC-DC converter for LED drivers. *IEEE Trans. Power Electron.* 29, 5392-5398 (2014).

(3) Regarding the optimal serial connection number (n), we predict that the peak power efficiency (PE) for PnP-QLED occurs at $n=37-38$. However, due to constraints related to the device's area, achieving this specific serial connection number is currently not feasible in our lab. Nevertheless, by examining the variations in device efficiency at $n=26, 28$, and 30 , as illustrated in Figure R6 (Supplementary Figure S19 in the revised manuscript), we observe that the trends align with our anticipated patterns. This is sufficient evidence to support the notion that efficient control of the serial connection number (n) is achievable for PnP-QLED.

Figure R6 a, b, c and d, $L-V_{RMS}$, $PE-V_{RMS}$, $CE-V_{RMS}$ and $EQE-V_{RMS}$ characteristic curves of $(\text{PnP-QLED})_n$ (with $n=26, 28, 30$), respectively. (reprinted from Supplementary Figure S19 of the revised manuscript)

(4) Regarding your concern about the stability of the connected devices, we already measured the lifetime of the AC-QLED driven by 50 Hz square wave AC, which shows a 2-fold longer lifetime than that of DC-QLED, as shown in Figure R7a (Fig. 2d-e in the revised manuscript). To further showcase the lifetime advantage of the connected device under sinusoidal AC, we measured the lifetime of PnP-QLED. As evident from Figure R7b (Supplementary Figure S15 in the revised manuscript), at an initial brightness of $10,020 \text{ cd/m}^2$, the T_{95} lifetime of PnP-QLED is 160 h, which is 1.5 times longer than that of DC-QLED. The results confirm the capability of (PnP-QLED)_n to achieve a prolonged operational life as an illumination device.

To clarify this point, a few sentences had been added to the revised manuscript, as follows (page 12, line 287-292 in the revised manuscript): “Furthermore, as shown in Supplementary Fig. S15, the T_{95} lifetime (initial average luminance of $10,020 \text{ cd/m}^2$) of the PnP-QLED under sinusoidal AC driving reaches 160 h, which is 1.5-fold longer than that of the DC-QLED. The results confirm that the PnP-QLED is capable of achieving both high efficiency and prolonged lifetime when operated under household AC electricity.”

Figure R7 a, The driving conditions for DC-QLED and AC-QLED for lifetime testing and the lifetime curves of DC-QLED and AC-QLED under $9,810 \text{ cd/m}^2$ and $10,020 \text{ cd/m}^2$, respectively. (reprinted from Fig.2d-e of the revised manuscript) **b**, The

driving conditions for PnP-QLED for lifetime testing and the lifetime curves of the PnP-QLED under 10,020 cd/m². (reprinted from Supplementary Figure S15 of the revised manuscript)

Comment #3: In Figure 4 and Figure S17, there seems to be a consistent, slight depression of Power Efficiency, Current Efficiency, and EQE around 175V in (PnP-QLED)₃₀. For (PnP-QLED)₂₆, (PnP-QLED)₂₈ case, there was a slight increase in PE, CE, and EQE at around 175V, whereas for (PnP-QLED)₃₀, it showed a decrease in such regions. However, there seems to be insufficient explanation regarding this matter. It is suggested that the authors clarify whether this is due to experimental error or consistent phenomena and provide an explanation regarding this matter.

Response #3: Thank you very much for your careful observation and pointing out this problem. The sudden drop observed in the efficiency of (PnP-QLED)₃₀ at 175 V is due to the instability of light emission from a specific QLED, which is a normal experimental error. However, this does not affect our analysis of the overall device behavior.

To investigate the randomness of this inflection point, we re-prepared (PnP-QLED)₃₀ and retested the device efficiency. As shown in the Figure R8, we found that its variation pattern still aligns with the trends observed in (PnP-QLED)₂₆ and (PnP-QLED)₂₈ in Figure R9 (Fig. 4c and Supplementary Figure S19b in the revised manuscript). We have already corrected these random experimental errors in the Fig. 4c and original Figure S17 (now Supplementary Figure S19 in Supplementary Information).

Figure R8 The PE of (PnP-QLED)₃₀ as a function of driving voltage. **a**, PE- V_{RMS} characterization. **b**, J - T - V characterization and P - T characterization at different V_{RMS} AC electricity.

Figure R9 a, The PE of (PnP-QLED)₃₀ as a function of n under the driving of 220 V/50 Hz AC electricity. (reprinted from Fig. 4c of the revised manuscript) **b**, $PE-V_{RMS}$ characteristic curves of (PnP-QLED) _{n} (with $n=26, 28, 30$). (reprinted from Supplementary Figure S19b of the revised manuscript)

Comment #4: *On page 2, the authors claim that removing the DC drivers and AC-DC converters results in energy losses and increased power consumption in the conversion process. Although we can agree on the fact that removing these complicated back-end circuits is beneficial in securing energy efficiency in general, given that there is noticeable energy loss in negative driving cycles, it can be suggested to add data regarding the comparison of efficiency between conventional household electronics (with circuitry) and this proposed work (without circuitry).*

Response #4: Thank you for your suggestions. We address each of your comments point-by-point below.

- (1) Regarding your concern on noticeable energy loss in negative driving cycles (Figure R10a-c), we would like to clarify that these losses only exist at low current and low brightness level ($<100 \text{ cd/m}^2$). At applicable brightness levels (>1000 or even 10000 cd/m^2 for lighting application), the losses are quite minimal and can be negligible, as can be clearly observed from Figure R10a-c (Fig. 2c and Supplementary Figure S10 in the revised manuscript).
- (2) For DC driving, the AC-DC conversion and DC-DC regulation circuitry typically consume 10~20% input energy¹⁻⁵, which is significantly higher than the energy losses ($<1\%$) in our PnP-QLED, as shown in Figure R10d-e (Supplementary Figure S11 in the revised manuscript).

To clarify this point, a few sentences had been added to the revised manuscript, as follows (page 9, line 232-235 in the revised manuscript): “As illustrated in Supplementary Fig. S11, the energy loss of the AC-QLED accounts for less than 1% under high voltage or luminance, significantly lower than the energy loss in switching

power supplies (AC-DC converters and DC-DC driver circuits), which typically ranges from 10% to 20%⁹⁻¹³,

Figure R10 AC-QLED energy loss under different AC driving cycles. a, b and c, EQE-J, CE-J and PE-L characteristic curves of devices under positive or negative driving cycles, respectively. (reprinted from Fig. 2c and Supplementary Figure S10 of the revised manuscript). The energy loss of AC-QLED is presented during positive and negative driving cycles and can be calculated by $\Delta P = P_{AC} - P_S$, where P_{AC} is the power of AC-QLED during positive or negative driving cycles, and P_S is the power of a single QLED (T-QLED or B-QLED). d and e, The $\Delta P-V$ and $\Delta P/P_{AC}-V$ characteristic curves of AC-QLED under positive or negative driving cycles, respectively. (reprinted from Supplementary Figure S11 of the revised manuscript)

References:

1. Chiu, H.-J., Lo Yu-Kang, Chen, J.-T., Cheng, S.-J., Lin, C.-Y. & Mou, S.-C. A high-efficiency dimmable LED driver for low-power lighting applications. *IEEE*

Trans. Ind. Electron. 57, 735-743 (2010).

2. Yu, W., Lai, J.-S., Ma, H. & Zheng, C. High-efficiency DC-DC converter with twin bus for dimmable LED lighting. *IEEE Trans. Power Electron.* 26, 2095-2100 (2011).
3. Pollock, A., Pollock, H. & Pollock, C. High efficiency LED power supply. *IEEE J. Emerg. Sel. Topics Power Electron.* 3, 617-623 (2015).
4. Zhang, F., Ni, J. & Yu, Y. High power factor AC-DC LED driver with film capacitors. *IEEE Trans. Power Electron.* 28, 4831-4840 (2013).
5. Malcovati, P., Belloni, M., Gozzini, F., Bazzani, C. & Baschiroto, A. A 0.18- μm CMOS, 91%-efficiency, 2-A scalable buck-boost DC-DC converter for LED drivers. *IEEE Trans. Power Electron.* 29, 5392-5398 (2014).

Comment #5: In Figure 2 and Figure S10, there is a considerable difference between losses (EQE, CE, PE) in negative driving conditions compared to positive driving conditions. Although the article mainly focuses on the high current range where losses can be neglected, it seems like more explanation can be suggested of why there is a deviation of loss between positive/negative driving conditions.

Response #5: Thank you for your suggestions. The difference of the efficiency losses between negative and positive driving cycles is due to the difference of the reverse current between B-QLED and T-QLED [Figure R11 (Fig.2b in the revised manuscript)], as we have discussed in the manuscript.

To further clarify this point, a few sentences had been added to the revised manuscript, as follows. (page 9-10, line 218-227 in the revised manuscript): “At low current or low brightness levels, the reverse current cannot be neglected, as it greatly affects the efficiency losses of the AC-QLED. As shown in Fig. 2c, it can be observed that the efficiency losses of AC-QLED under positive and negative driving cycles (green shaded areas) are not equal. Due to the rougher surface induced by the thick bottom layers, the reverse current J_{R_T} (from the T-QLED) during the negative driving cycles is much higher than the reverse current J_{R_B} (from the B-QLED) during the positive driving cycles, as illustrated in Fig. 2b. Consequently, under negative driving cycles, the energy losses in the AC-QLED are higher compared to those under positive driving cycles. However, for high currents or high brightness situations, because the forward current is significantly higher than the reverse current, the impact of reverse current on the efficiency of AC-QLED becomes negligible.”

Figure R11 J - V characteristics of T-QLED, B-QLED, and AC-QLED in positive driving cycles (top) and negative driving cycles (bottom). (reprinted from Fig. 2b of the revised manuscript)

Comment #6: *The authors show optical simulation results of the tandem QLED in Fig. 1c, Fig. S4, and Fig. S5 to optimize the thickness of the layers of the device. It would be beneficial to have a detailed description of the simulation conditions and methods, along with an extended discussion of the results. On page 5 of the main text, the author claims that the size of one cavity is impacted by the size of another cavity. As a result, the tandem device was conceived as two Fabry-Pérot resonant cavities stacked vertically. However, the author presented simulated data without accompanying explanations. Hence, a comprehensive clarification of the simulation principles and results is necessary. Furthermore, since these results are purely from simulations, empirical data would be provided to validate these findings.*

Response #6: Thank you for your good suggestions. We address each of your comments point-by-point below.

- (1) Following your suggestion, we had provided the detailed simulation method in the Supplementary Information and the refractive index used for simulation.
- (2) Regarding the explanations to the simulated results, we would like to point out that optical simulation is not the key point of our paper. We treated it as a tool to

optimize our tandem QLEDs. Since the simulated results are in good agreement with the experimental data, we prefer not to add more explanations so that the readers can focus on our main point on household AC electricity directly driven PnP-QLED.

(3) Regarding the validation of the simulated data, actually we have already presented a comparison between simulation and experimental results to validate the accuracy of the simulation. As shown in the Figure R12 (Supplementary Figure S4h and Supplementary Figure S9g in the revised manuscript), the trends in EQE/γ (obtained by simulation) for both QLED and stacked QLED, are consistent with the experimental data, confirming the precision of our optical simulation.

To clarify this point, a few sentences had been added to the revised manuscript, as follows. (Page 5, line 137-142: in the revised manuscript): “The optical performance of the devices can be simulated by using a dipole radiation model⁶⁸⁻⁷¹. The simulation methods and refractive index used for simulation are detailed in Supplementary Simulation Method. To validate the accuracy of the simulation, we compare the simulated EQE/γ with the measured EQE. As shown in Supplementary Fig. S4h, the trend of the EQE/γ of the QLED is in good agreement with the measured EQE, confirming the accuracy and reliability of our simulated methods.”

Figure R12 a, Comparison of experimental (blue curve) and simulation results (red curve) for EQE of QLED as a function of ZMO thickness. (reprinted from Supplementary Figure S4h of the revised manuscript) **b**, Comparison of experimental results (blue curve) and simulation results (red curve) for EQE of S-QLED as a function of IZO thickness. (reprinted from Supplementary Figure S9g of the revised manuscript)

Comment #7: *In Figure 1e, the luminance data for B-QLED and T-QLED ceases beyond 8V, whereas S-QLED continues to persist beyond 16V. Is the absence of data beyond 8V indicative of device damage, or is there another underlying reason? If that's the case, at what voltage was the device's lifetime tested and driven in Figure 2d?*

Response #7: In Figure R13a (Fig. 1e in the revised manuscript), the B-QLED and T-QLED are individual QLED devices, and thus a driving voltage of 8 V is high enough for them. The S-QLED is the tandem device, which consists of a B-QLED and a T-QLED connected in series, and therefore, its maximum driving voltage is equal to the summation of those of the B- and T-QLED, which is exactly 16 V.

For lifetime testing, the devices were stressed by a constant current rather than voltage. The detailed stressing conditions are shown in Figure R13b-c (Fig. 2d-e in the revised manuscript).

Figure R13 a, The J - V - L characteristics of B-QLED, T-QLED, and S-QLED. (reprinted from Fig. 1e of the revised manuscript) **b,** the driving conditions for S-QLED and AC-QLED for lifetime testing. **c,** the lifetime curves of the S-QLED (red curve) and AC-QLED (blue curve). (reprinted from Fig. 2d-e of the revised manuscript)

Comment #8: In Figure S17, the legends in Figures a, b, c are labeled inconsistently. Figure S17 a, c's legend is labeled "the (PnP-QLED)₂₆," where for Figure S17 b's legend is labeled, "(PnP-QLED)₂₆." To prevent confusion, correction is suggested.

Response #8: Thank you for pointing out this mistake. We have carefully examined the whole paper and make necessary corrections.

Comment #9: Some scale bars are missing in the photographs of Figures.

Response #9: Thank you for pointing out this mistake. We have carefully examined the whole paper and made necessary corrections.

We hope our responses/revisions satisfactorily address all your concerns. Once again, we thank you for your constructive and helpful suggestions!

Response to Reviewer #3

General comment: *The authors reported their work on household AC electricity plug-and-play quantum-dot light-emitting diodes. They made high-performance tandem QLEDs, which can be operated at both negative and positive cycles of the AC voltage, by using a transparent and conductive IZO as an intermediate electrode resulting in high EQEs of 22.99%. The idea is good and the experimental data are reliable. However, there are a few issues to be addressed before it could be published in this journal.*

Our response: Thank you for your efforts in reviewing this manuscript. We sincerely appreciate your positive comments on our idea and our reliable data. Also, your constructive suggestions are very helpful, which greatly help us to improve the quality of this paper.

Comment #1: *Although it would be cost-effective that their tandem QLEDs can be directly driven by 220/110 V AC electricity without the need for complicated back-end circuits, the AC-QLEDs, similar to some other AC driven lighting sources, cannot provide with real continuous light emission with steady light output. Actually, in-house lighting needs a steady light output for eye health. How to provide with steady light output is a big challenge to this AC-QLEDs. The authors should find a way to resolve this problem. What is the dark interval time when the voltage was changed from the positive to the negative?*

Response #1: We fully agree that the AC driven light sources cannot provide real continuous light emission with steady light output. The unstable light output can result in undesired luminance flickering, which is a common issue **that can be observed in all AC light sources** such as the incandescent bulbs or fluorescent lamps. For almost all light sources, there is usually a trade-off between lamp cost and luminance flickering.

Figure R14a (Supplementary Figure S24a in the revised manuscript) shows the time-resolved electroluminescence (TrEL) of (PnP-QLED)₁₆ and (PnP-QLED)₃₀ driven by household 220 V power supply. Indeed, at low voltage, the device cannot be turned-on and there is no light output. The dark time of (PnP-QLED)₃₀ are 1.56 ms in a AC cycle of 20 ms. By reducing the number of QLEDs used for connection, the turn-on voltage can be reduced, and thus the dark time can be decreased. For example, the dark time can be reduced to 0.38 ms for (PnP-QLED)₁₆.

For our (PnP-QLED)_n, the luminance flickering can be mitigated by improving the continuity of the light output. To this end, we propose a method involving the

parallel connection of two $(\text{PnP-QLED})_n$ devices, with a modification of the AC electricity phase for one $(\text{PnP-QLED})_n$ device, as illustrated in Figure R14b (Supplementary Figure S24b in the revised manuscript). The modification of the phase of the AC electricity can simply be achieved by using a capacitive delayer. By delaying 5 ms, the phase of the AC is altered by 90° . Under household AC electricity driving, the first $(\text{PnP-QLED})_n$ is rapidly powered on, while the second $(\text{PnP-QLED})_n$ is powered on 5 ms later. When the first $(\text{PnP-QLED})_n$ exhibit the maximum EL, the second $(\text{PnP-QLED})_n$ exhibit the minimum (dark) EL, as shown in Figure R14c (Fig 4f in the revised manuscript). Therefore, the total EL from both $(\text{PnP-QLED})_n$ is relatively stable and continuous, thus significantly reducing the luminance flickering.

To clarify this point, a few sentences had been added to the revised manuscript, as follows. (Page 15-16, line 353-377 in the revised manuscript): “To investigate whether there is a luminance flickering that is commonly observed in AC driven light sources such as incandescent bulbs or fluorescent lamps, the time-resolved electroluminescence (TrEL) of $(\text{PnP-QLED})_{30}$ under household AC electricity driving was measured. As shown in Fig. 4f, the driving voltage varies in a sinusoidal manner, and at low driving voltage, the device cannot be switched on and therefore there is no light output. For example, for the $(\text{PnP-QLED})_{30}$ driven by a 220 V household power supply, there are 1.56 ms in an AC cycle (20 ms) that the device is in a dark state. Because the light output during an AC cycle is not continuous and steady, the human eye perceives luminance flickering. The luminance flickering can be mitigated by improving the continuity of light output. By reducing the value of n , the turn-on voltage of $(\text{PnP-QLED})_n$ can be reduced, and thus the duration of the dark period can be decreased. For example, when the n decreases from 30 to 16, the dark period of $(\text{PnP-QLED})_n$ effectively decreases from 1.56 to 0.38 ms (Supplementary Fig. S24a), thereby leading to the suppression of luminance flickering. To achieve a more stable light output and further reduce luminance flickering, we connected two $(\text{PnP-QLED})_{30}$ devices in parallel and modified the phase of the AC electricity for one $(\text{PnP-QLED})_{30}$ device, as schematically illustrated in Supplementary Fig. S24b. The modification of the phase of the AC electricity can simply be achieved by using a capacitive delayer. By delaying 5 ms, the phase of the AC is altered by 90° . Under household AC electricity driving, the first $(\text{PnP-QLED})_{30}$ is rapidly powered on, while the second $(\text{PnP-QLED})_{30}$ is powered on 5 ms later. Due to the 90° phase difference of the AC driving, the EL of both $(\text{PnP-QLED})_{30}$ is different at a given time. For example, when the first $(\text{PnP-QLED})_{30}$ exhibit the maximum EL, the second $(\text{PnP-QLED})_{30}$ exhibit the minimum (dark) EL, as shown in Fig. 4f. Therefore, the total EL from both $(\text{PnP-QLED})_{30}$ is relatively stable and continuous, thus significantly

reducing the luminance flickering.”

Figure R14. **a**, The TrEL of $(PnP-QLED)_{16}$ and $(PnP-QLED)_{30}$ directly driven by 220 V/ 50 Hz household AC power supply. **b**, Schematic circuit diagram of two $(PnP-QLED)_n$ under 220 V/50 Hz AC driving (reprinted from Supplementary Figure S24 of the revised manuscript). **c**, The TrEL $(PnP-QLED)_{30}$ directly driven by 220 V/ 50 Hz household AC power supply with different phase difference. (reprinted from Fig. 4f of the revised manuscript).

Comment #2: The authors mentioned regarding Movie 3 that “the flicker is due to the mismatch of the QLED lighting frequency and the frame captured frequency of the camera. The emission is quite stable observed by eyes”. However, it needs measurement data to prove the statement. Therefore, it is suggested the authors should show the relationship between light output vs. time under $V(RMS) = 220\text{ V}$ with a chart similar to that in Figures S19 and S20.

Response #2: Thank you for your valuable suggestions. The relationship between light output vs. time under 220V/50 Hz driving is shown in Figure R14c (Fig. 4f in the revised manuscript). Although it is difficult to detect the luminance flicker with human eye, it is clearly visible in the measurement data. The dark period at low voltage is the main cause of uncontinuous light output and luminance flicker. To eliminate the dark period, we have proposed a method as we specifically discussed in **Response #1**.

Comment #3: Fig S2a: The $V-AC$ on the top should be put in one line. Fig. S2c: Since Electrodes A and B contact to the same charge generation layer, the energy

levels of both electrodes relating to the HOMO of the charge generating layer should be the same.

Response #3: Thank you for your helpful suggestion. We have already corrected the errors in the original Fig. S2a (now Supplementary Figure S2a in Supplementary Information).

Comment #4: Fig. S4: In the annotation, “b, Simulated outcoupling efficiency of b, B-QLED, ...ITO”: It might be better to put the chart numbers in the front of the sentence. How about “b, c, and d, Simulated outcoupling efficiency of B-QLED, T-QLED, and S-QLED, respectively, as a function of ...ITO”? “B-QLED” should be “B-QLED”.

Response #4: Thank you for your good suggestion. Following your suggestion, we have modified the annotation of original Fig. S4 (now Supplementary Figure S5 in Supplementary Information).

Comment #5: Fig. S4: In the annotation: “Power fraction of each mode as a fraction of the ITO thickness in red e, B-QLED....”, What is the meaning of “in red e”? Red color in the chart is only the air mode of the power fraction.

Response #5: Sorry for the confusion. We have refined our expression of the annotation in original Fig. S4 and Fig. S5 (now Supplementary Figure S5 and Supplementary Figure S6 in Supplementary Information).

Comment #6: Fig. S5: “80 nm, 20 nm, 32%” is good for the B-QLED, “25 nm, 110 nm, 32%” is good for the T-QLED. How do you conclude that 30 nm, 90 nm is good for the S-QLED?

Response #6: Sorry for the confusion. In original Fig. S5 (now Supplementary Figure S6 in Supplementary Information), the B-QLED and T-QLED are optimized separately. Thus, although “80 nm, 20 nm, 32%” is the best condition for the B-QLED, it is not the best condition for the T-QLED. In S-QLED shown in Figure R15a (Fig. 1b in the revised manuscript), we consider the maximum total output from both B-QLED and T-QLED, rather than the maximum output from individual QLED. Therefore, the corresponding device structure needs to ensure that the sum of EQE/γ of B-QLED and T-QLED is maximized, rather than maximizing the EQE/γ of one of the devices. Therefore, as shown in Figure R15b (Fig. 1c in the revised manuscript), the maximum sum of OCEs of B-QLED and T-QLED corresponds to thicknesses of 30 nm for IZO and 90 nm for ZMO_T.

Figure R15 a, Schematic optical model of a tandem QLED. To achieve constructive wide-angle interference, the length of both cavities should be simultaneously optimized. **b**, Simulated OCE of S-QLED as a function of the thickness of ZMO_T and IZO. (reprinted from Fig. 1b-c of the revised manuscript).

Comment #7: Fig. S6: In order to understand the dynamic spin-coating, it is suggested to show the static spin-coating condition in this work.

Response #7: Thank you for your suggestion. Figure R16a-b (Supplementary Figure S7a-b in the revised manuscript) provides a schematic diagram illustrating the process of both static and dynamic spin-coating operations.

Figure R16 a and b, The diagram of static spin-coating and the dynamic spin-coating, respectively. (reprinted from Supplementary Figure S10a-b of the revised manuscript).

Comment #8: Fig. S7: On the bottom-right, “B-QLED” should be “T-QLED”.

Response #8: Thank you for your careful reading and pointing out our mistake. We have already corrected the errors in the original Fig. S7 (now Supplementary Figure S8 in Supplementary Information).

Comment #9: Fig. S8: 1) The thickness of D1 is also critical to the outcoupling of the light. However, there is no any discussion on D1. 2) If caption “ZMO thickness” is

used, “Current Density” should be changed as “Current density”. This “capital initial letter” unconsistence can also be seen in the other figures, such as Figs. S9, S10, S17.

Response #9: Thank you for your helpful suggestion. Indeed, D_1 does affect the device's outcoupling efficiency. Based on the simulation results shown in Figure R17 (Supplementary Figure S4d in the revised manuscript) and our previous extensive research on the basic device, we concluded that the outcoupling efficiency of the device is highest when the thickness of D_1 is 60 nm, leading to optimal performance (EQE) of the device. Therefore, in the basic device, we have already optimized the thickness of D_1 to achieve the best results.

To clarify this point, in “Supplementary Information”, a few sentences had been added, as follows (Supplementary Information page 23): “Our previous work³⁰ indicates that a QLED, at the optimal resonant cavity (D_1) thickness of 60 nm, can exhibit the highest performance, confirming the accuracy of the optical simulations. Therefore, during the optimization process of the OCEs of the stacked device, the D_1 (CBP and MoO_3) was fixed at 60 nm.”

Figure R17 The outcoupling efficiency for QLED as a function of the thickness of D_1 . (reprinted from Supplementary Figure S4d of the revised manuscript)

Also, following your suggestion, we have corrected the captions in the figures (Supplementary Figure S9-10, S19 in Supplementary Information).

We hope our responses/revisions satisfactorily address all your concerns. Once again, we thank you for your constructive and helpful suggestions!

REVIEWERS' COMMENTS

Reviewer #1 (Remarks to the Author):

All the comments are well addressed in the revision and the responses are convincing. I recommend its publication.

Reviewer #2 (Remarks to the Author):

The concerns I raised regarding the novelty of the work seem to be properly addressed with the additional results and explanation. The reviewer believes that the work can be suitable for publication without further modification.